# Maturation and specialization of group 2 innate lymphoid cells through the lung-gut axis

Min Zhao[1,4], Fei Shao[1,2,4], Dou Yu[1,2,4], Jiaqi Zhang[1,2], Zhen Liu[1,2], Jiangwen Ma[1,2], Pengyan Xia ®[3] ✉ & Shuo Wang ®[1,2] ✉

Innate lymphoid cells (ILC) are abundant in mucosal tissues. They serve critical functions in anti-pathogen response and tissue homeostasis. However, the heterogenous composition of ILCs in mucosal sites and their various maturation trajectories are less well known. In this study, we characterize ILC types and functions from both the lung and the small intestine, and identify their tissue-specific markers. We find that ILC2s residing in the lung express CCR2, whereas intestinal ILC2s express CCR4. Through the use of CCR2 and CCR4 reporter mice, we show that ILC2s undergo translocation via the lung-gut axis upon IL-33 treatment. This trajectory of ILC2s is also observed at the postnatal stage. Allergen-induced activation of lung ILC2s affects the homeostasis of gut ILC2s. Together, our findings implicate that ILCs display tissue-specific features in both the lung and gut, and ILC2s mature along the lung-gut axis in particular homeostatic and inflammatory conditions.

The mucosal immune system is regionally-restricted with specialized features separate from the systemic immune system. Innate lymphoid cells (ILCs) reside at mucosal surfaces and serve important functions in both anti-pathogenic response and tissue homeostasis[1]. Two major mucosal tissues where ILCs exert various functions include the lung and intestine. ILC1s have been found at increased levels in the lung in cases of influenza virus infection and accumulate in the gut of patients with Crohn's disease[2,3]. Lung ILC2s participate in pulmonary inflammation[4], whereas intestinal ILC2s contribute to anti-parasite response[5]. Few reports have studied the function of ILC3s in the lung. Intestinal ILC3s are well-known to have activities in anti-bacterial response, inflammation, and tissue homeostasis[6]. ILCregs are only observed in the intestine, serving as regulators of colitis[7]. In general, little is known about the differential functions and tissue-specific markers of ILCs within the lung and gut.

Group 2 innate lymphoid cells (ILC2s) are abundant in the mucosal surface of both the lung and the gut. Under steady-state conditions, ILC2s reside in mucosal sites[8]. Upon stimulation by

cytokines or external antigens, ILC2s undergo dynamic changes and circulate throughout the immune system. For example, fungal allergen-induced IL-33 enables the egress of ILC2s from bone marrow (BM) and subsequent trafficking to the lung[9]. Helminth infection or IL-25 stimulates inflammatory ILC2s (iILC2s) to migrate from the intestine to the lung[10]. Other studies have shown that iILC2s are able to be recruited to mesenteric lymph nodes and thus circulated throughout the peripheral blood[11]. In sum, the dynamic changes of ILC2s are associated with the processes of mucosal-associated diseases.

Although ILCs are functionally parallel to T helper (Th) cells, ILCs undergo their unique developmental steps in BM and peripheral tissues[12]. The common progenitors of ILCs have been identified in BM and specific precursors for each group of ILC observed. Recently, liver ILC1s were found to develop locally depending on IFN-γ[13]. As for ILC2s, ILC2Ps localize in the BM and develop into ILC2s depending on TGF-β and IL-33[14,15]. ILC3Ps have been identified in the BM of mice and in the tonsils and intestines of humans[16]. Through these cases, we know that the commitment of ILCs occurs in BM and peripheral tissues. However,

---

[1]CAS Key Laboratory of Pathogen Microbiology and Immunology, Institute of Microbiology, Chinese Academy of Sciences, 100101 Beijing, China. [2]University of Chinese Academy of Sciences, Beijing 100049, China. [3]Department of Immunology, School of Basic Medical Sciences, NHC Key Laboratory of Medical Immunology, Peking University, Beijing, China. [4]These authors contributed equally: Min Zhao, Fei Shao, Dou Yu. ✉e-mail: xiap@pku.edu.cn; wangshuo@im.ac.cn

the specialization of ILCs in various mucosal sites and their maturation trajectories along the mucosal tissues are largely unknown.

In this work, we generate expression profiles of ILCs in both the mouse lung and intestine through the use of single-cell RNA sequencing. Eleven subsets of ILCs are identified as well as tissue-specific markers for each of these groups. Our findings imply that lung ILC2s tend to express CCR2, whereas intestinal ILC2s express CCR4. Moreover, ILC2s are able to undergo maturation through the lung-gut axis. During the development of ILC2s at postnatal stage, triggering of allergic asthma induced hyperactivation of ILC2s in the lung and gut. Our study characterizes the specialized features of ILCs in the lung and gut and highlights the importance of the lung-gut axis in the maturation process of ILC2s.

## Results

### ILC subgroups in the small intestine and lung

In order to identify the subpopulations of ILCs, we performed single-cell (sc) RNA sequencing on ILCs sorted from the mouse small intestine and lung (Supplementary Figure 1a). After excluding contaminates and low-quality cells, we identified 11 distinct subgroups of ILCs in the small intestine (SI) and lung, including ILC1 (SI, Lung), ILC2 (SI-A, SI-B, Lung-A, Lung-B), ILC3 (SI-A, SI-B, SI-C), and ILCreg (Fig. 1a, b). Notably, these ILCs demonstrated high spatial heterogeneity. ILC type differences between the lung and gut were found to be much larger than between ILC subsets (Fig. 1c). We mapped each ILC cluster to its respective group according to previously reported ILC markers, and none of them express NK cell marker, *Eomes* (Fig. 1d and Supplementary Fig. 1b)[17]. Signature genes present in each cluster were determined, which indicated the specialized functions of the particular group (Fig. 1e). After further analysis involving ILC cluster frequency (Fig. 1f), we found that the distribution of each cluster was highly restricted to either the lung or the gut.

### Tissue-specific markers of ILCs in the small intestine and lung

Next, we compared differentially-expressed genes (DEGs) in each ILC group to identify their tissue-specific genes (Fig. 2a–c). Lung ILC1s expressed high levels of *Anxa1* and *Anxa2*, which are members of calcium-regulated membrane-binding proteins–this indicates a potential function of calcium in the regulation of lung ILC1s (Fig. 2a)[18]. Intestinal ILC1s were unique in their expression of *Gpr55* and *Cxcr6*, suggesting the regulation of GPCR signaling within them. As for ILC2s, expression levels of IL-33 receptor (*Il1rl1*), Leukotriene B4 receptor (*Ltb4r1*), and *Ccr2* were significantly higher in lung ILC2s compared to intestinal ILC2s (Fig. 2b). Conversely, intestinal ILC2s harbored elevated levels of IL-25 receptor (*Il17rb*), *Hilpda*, and *Ccr4*. Lung ILC3s occupied a small population with high levels of *Ccr6*, *Ccr7*, and *Cd4*, suggesting that NCR⁻ILC3s were the major subgroup of ILC3s present in the lung (Fig. 2c). The intestinal ILC3s expressed high levels of *Stat5*, *Cxcr6*, and *Ffar2*, which is a finding consistent with previous studies[19]. Gene Ontology (GO) analysis illustrated that ILCs in the lung tended to express genes related to cell adhesion, the extracellular matrix, and cell differentiation. Intestinal ILCs harbored high levels of genes associated with kinase activity, cytokine production, and metabolism (Fig. 2d–f and Supplementary Data 1).

We then selected tissue-specific markers for the pan-ILCs. We found that mRNA levels of *Cxcr6*, *Stat5a*, and *Rora* were much higher in gut ILCs compared to lung ILCs. The expression of *Anxa1*, *Ly6a*, and *S100a4* was more specific to lung ILCs (Fig. 2g). After determining markers for these overarching groups, we moved to identify tissue-specific markers for ILC subgroups. *Gpr55* was specifically expressed in intestinal ILC1s while *Sell* expression was unique to lung ILC1s. The intestinal ILC2s expressed *Ccr4*, whereas lung ILC2s contained high levels of *Ccr2*. *Ffar2*, a metabolism-associated regulator of ILC3s[19], was only expressed in the gut ILC3s. *Cd36* was identified as a lung-specific ILC3 marker. Thus, tissue-specific markers of ILCs imply their local

environments endow them unique function. These markers might help us to trace ILC location and analyze their dynamic regulation between the lung and gut.

### CCR2 and CCR4 serve as tissue-specific markers of ILC2 in the lung and gut, respectively

To identify tissue-specific markers of ILC2, we further analyzed the single-cell RNA sequencing data. All ILC2 subsets expressed the signature genes of ILC2s, including *Gata3*, *Ly6a*, *Il1rl1* (ST2), and *Klrg1* (Fig. 3a). They did not express the marker genes of ILC2 precursors (Supplementary Fig. 2a)[20]. We found that *Ccr2* and *Ccr4* expression was restricted to lung ILC2s and gut ILC2s, respectively (Fig. 3a). Notably, the expression levels of *Gata3* and *Klrg1*, which were correlated with the maturation of ILC2s[21], gradually increased in ILC2 subsets from the lung to the gut. We made use of pseudotime analysis to order the maturation process of ILC2 subsets (Fig. 3b). Intriguingly, the ILC2-lung-A subgroup located mainly at the beginning of the trajectory. The ILC2-lung-B subset was distributed at the origin and a developed branch 1 of ILC2s. There are two branches of ILC2s – branch 2 is considered more mature than branch 1. However, few intestine ILC2s belonged to the start site of the trajectory. ILC2-SI-A belonged to two branches of ILC2s and ILC2-SI-B only occupied the final stage of maturation. Lung ILC2s were distributed at the beginning and branch 1 of the trajectory. Intestinal ILC2s belonged to the differentiated branches (Supplementary Fig. 2b). Along the trajectory, the expression levels of mature marker genes of ILC2s (*Gata3* and *Klrg1*) apparently increased and genes indicating less mature of ILC2s (*Ly6a and Il1rl1*) decreased (Supplementary Fig. 2c). We further analyzed the scRNA-sequencing data of ILC2s in various tissues. Lung ILC2s were less mature and closest to the ILC2s in BM (Supplementary Fig. 2d)[22].

In order to trace ILC2s in the lung and gut, we generated CCR2-mNeonGreen and CCR4-mNeonGreen reporter mice. We found that CCR2-mNeonGreen is specifically expressed on lung ILC2s and CCR4-mNeonGreen is specific for gut ILC2s (Fig. 3c, d and Supplementary Fig. 3a). We confirmed the expression of CCR2 on lung ILC2s and CCR4 on gut ILC2s of our reporter mice (Supplementary Fig. 3b). Then, we used IL-25 and IL-33 to stimulate ILC2s of the reporter mice in vivo. IL-25 is required for the induction of inflammatory ILC2s (iILC2s) in the lung[10], which are ST2⁻KLRG1^hi. We observed that iILC2s in the lung were CCR2 negative (Fig. 3e) and CCR4 positive (Fig. 3f), indicating that they might retain the mNeonGreen fluorescence signal of gut ILC2s. Additionally, ST2⁺KLRG1⁺ ILC2s in the lung were CCR2⁺CCR4⁻, suggesting that they were not from the gut (Fig. 3e, f). IL-33 treatment increased the level of CCR2⁺CCR4⁻ ILC2s in the lung (Fig. 3g, h). Gut ILC2s were mainly CCR2⁻CCR4⁺ upon stimulation by either IL-25 or IL-33 (Fig. 3e–h). Notably, a small proportion of gut ILC2s were CCR2-mNG⁺, but there were no CCR4-mNG⁺ILC2s in the lung after IL-33 treatment (Fig. 3g, h).

### ILC2s undergo translocation through the lung-gut axis

To understand the dynamic change of ILC2s in the lung and gut, we took advantage of CCR2-RFP; CCR4-mNeonGreen reporter mice (Fig. 4a). The majority of lung ILC2s were CCR2⁺ (RFP⁺), while most of the gut ILC2s were CCR4⁺ (mNeonGreen⁺) at steady state. Intriguingly, IL-33 was able to induce RFP⁺ILC2s in the gut (Fig. 4a)–this finding is consistent with previous data (Fig. 3g). We also observed RFP⁺mNeonGreen⁺ double positive ILC2s in the gut (Fig. 4b and Supplementary Fig. 3c), suggesting that these ILC2s may have underwent a transition from CCR2⁺ILC2s to CCR4⁺ILC2s. We further evaluated the frequency of RFP⁺ and/or mNeonGreen⁺ ILC2s after prolonged stimulation by IL-33. We found that the presence of RFP⁺mNeonGreen⁺ ILC2s gradually increased with the reduction of RFP⁺ single positive ILC2s (Fig. 4b). Notably, RFP⁺ILC2s and RFP⁺mNeonGreen⁺ILC2s were also observed in the gut of mice intranasally treated with IL-33 or house dust mite (HDM) (Supplementary Fig. 3d, e). We next analyze the

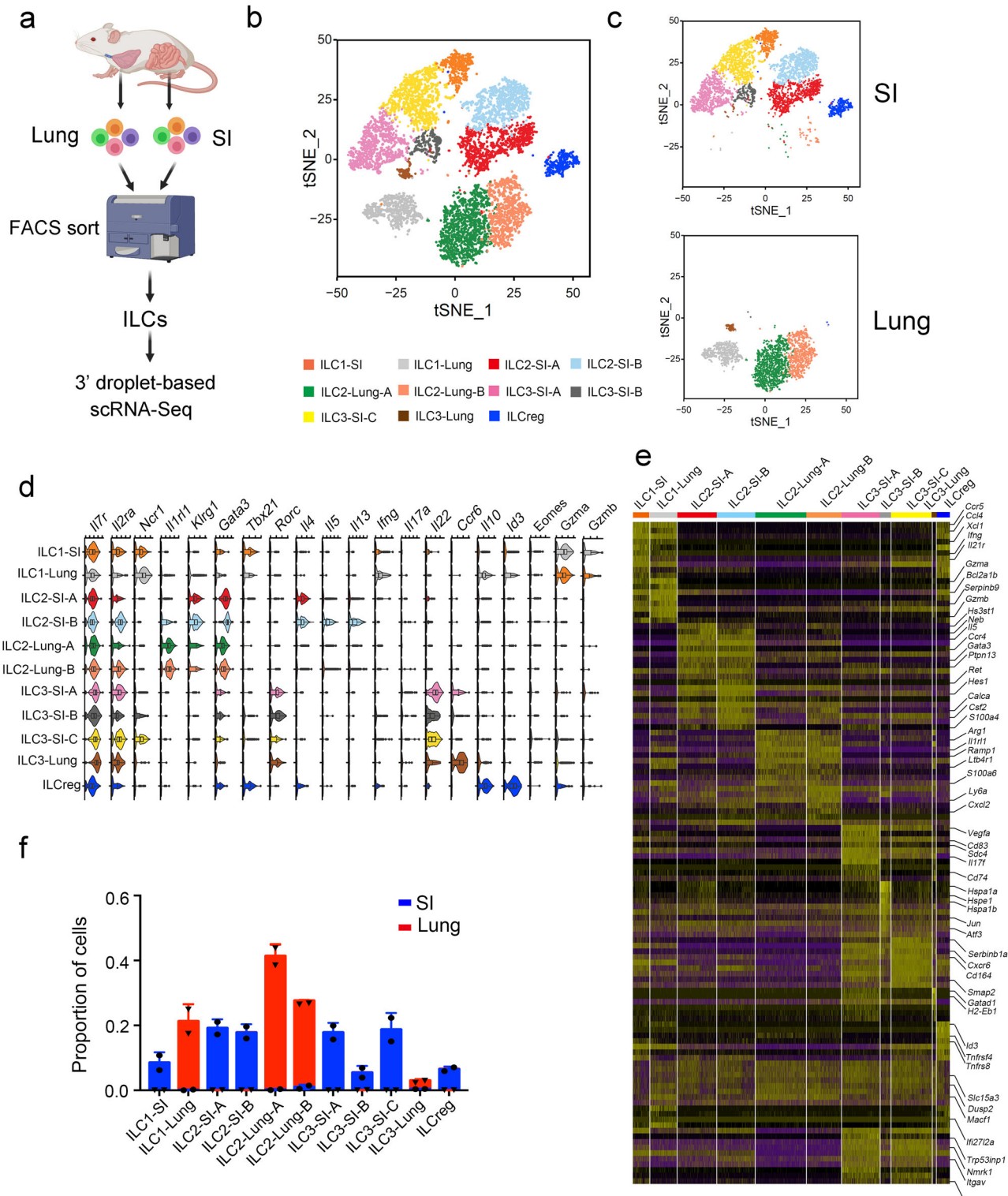

**Fig. 1 | Heterogeneity of ILCs in the lung and small intestine. a** Experimental design for scRNA sequencing of ILCs in the lung and small intestine. We isolated the ILCs (Lin⁻CD45⁺CD127⁺) from the lung and small intestine (SI) of wild type (WT) mice (Lin=CD3e,CD8a,CD19,CD11b,CD11c,Gr1,F4/80,Ter119) by flow cytometry. Isolated ILCs were subjected to 3' droplet-based scRNA sequencing. The mice share the similar distribution of ILCs from the same tissue and we pooled them together as one sample for scRNA-seq (*n* = 5 for each sample). Two biological repetitions were made for each sample. The schematic diagram was created with BioRender.com. **b** ILC subsets were determined and shown by t-Distributed stochastic neighbor embedding (tSNE). Low quality cells and contaminates were removed. Eleven ILC clusters were identified. **c** Distribution of identified ILC subsets in the lung and gut. **d** Expression levels of ILC signature genes were analyzed by violin plot. Box plots indicate median (middle line), 25th, 75th percentile (box) of the data and the maximum and minimum values as endpoints for the whiskers. *n* = 454 for ILC1-SI, 734 for ILC1-Lung, 1063 for ILC2-SI-A, 1033 for ILC2-SI-B, 1376 for ILC2-Lung-A, 944 for ILC2-Lung-B, 1036 for ILC3-SI-A, 292 for ILC3-SI-B, 1107 for ILC3-SI-C, 109 for ILC3-Lung, 363 for ILCreg. **e** Top ten signature genes for indicated ILC subtypes were shown in the heat map. **f** Proportions of each ILC cell subset of (**d**) in the lung and gut ILCs were calculated and shown as mean ± SD. Source data are provided as a Source Data file.

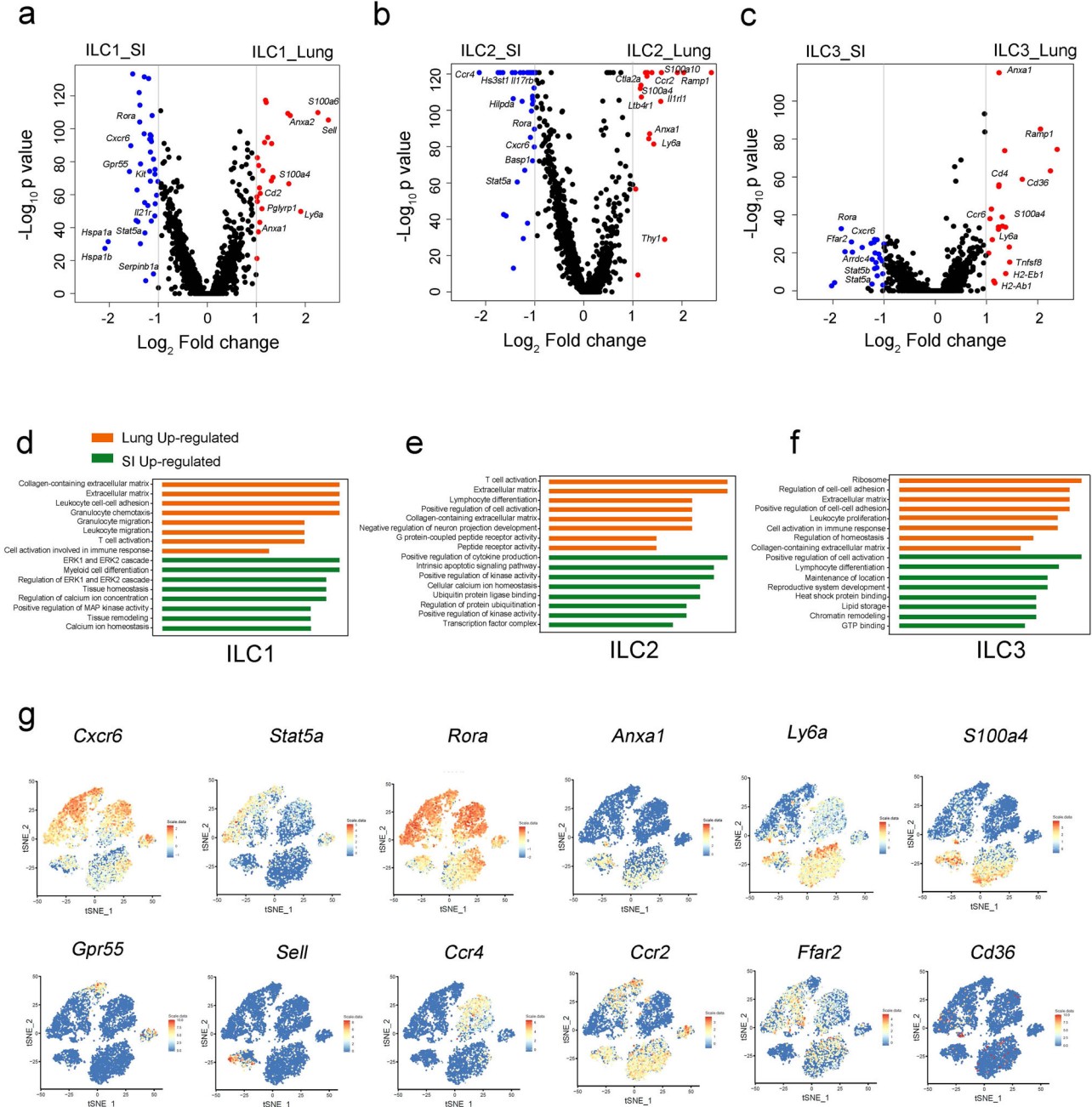

**Fig. 2 | Differentially expressed and tissue specific genes of ILCs in the lung and small intestine. a–c** Differential gene expression of ILC subsets in the lung and the gut was analyzed and shown in Volcano plot. Statistical analysis was performed by using two-tailed unpaired Student's *t*-test. Red dots represented the genes highly expressed in the lung, and blue dots represented the genes highly expressed in the gut. **d–f** GO analysis of differentially expressed genes in the lung and the gut. The orange bars showed the pathways that were enriched in the lung, and the green bars showed the pathways that were enriched in the small intestine. **g** Normalized relative expression (log(scaled UMI + 1)) levels of the selected tissue specific genes of ILCs displaying on a tSNE plot of Fig. 1b. *Cxcr6, Stat5a* and *Rora* were enriched in the intestine ILCs, and *Anxa1, Ly6a* and *S100a4* were highly expressed in the lung ILCs. Selected tissue specific genes for ILC subsets are: *Gpr55* for intestine ILC1s, *Sell* for lung ILC1s, *Ccr4* for intestine ILC2s, *Ccr2* for lung ILC2s, *Ffar2* for intestine ILC3s, and *Cd36* for lung ILC3s. Source data are provided as a Source Data file.

mRNA levels of *Ccr2* and *Ccr4* in ILC2s from these mice. Interestingly, RFP⁺ILC2s in the gut did not express *Ccr2* mRNA (Supplementary Fig. 3f). Moreover, IL-33 treatment was not able to induce *Ccr2* expression in gut ILC2s in vitro (Supplementary Fig. 3g), indicating that they just maintained the RFP protein from the lung. Along with evidence from pseudotime analysis, we speculated that lung ILC2s were experienced transition and adaption in the gut upon stimulation.

To further monitor the transition/maturation of ILC2s in the lung and gut, we transferred lung or gut ILC2s from IL-33-treated mice into immunodeficient mice. We found that lung ILC2s were able to give rise

to gut ILC2s. However, gut ILC2s failed to generate lung ILC2s or expressed *Ccr2* (Fig. 4c and Supplementary Fig. 3h, i). We further transferred ILC2 precursors into immunodeficient mice and subsequently stimulated these cells with IL-33 (Fig. 4d). Here, we determined that reconstruction of ILC2s occurred earlier in the lung than in the gut (Fig. 4d). The frequency of gut ILC2s gradually increased alongside that of lung ILC2s.

Next, we took advantages of tamoxifen-inducible Cre/ERT2 system for cell tracing. We delivered 4-Hydroxy-Tamoxifen (4-OHT) with liquid aerosol devices, which enable transient tracing of

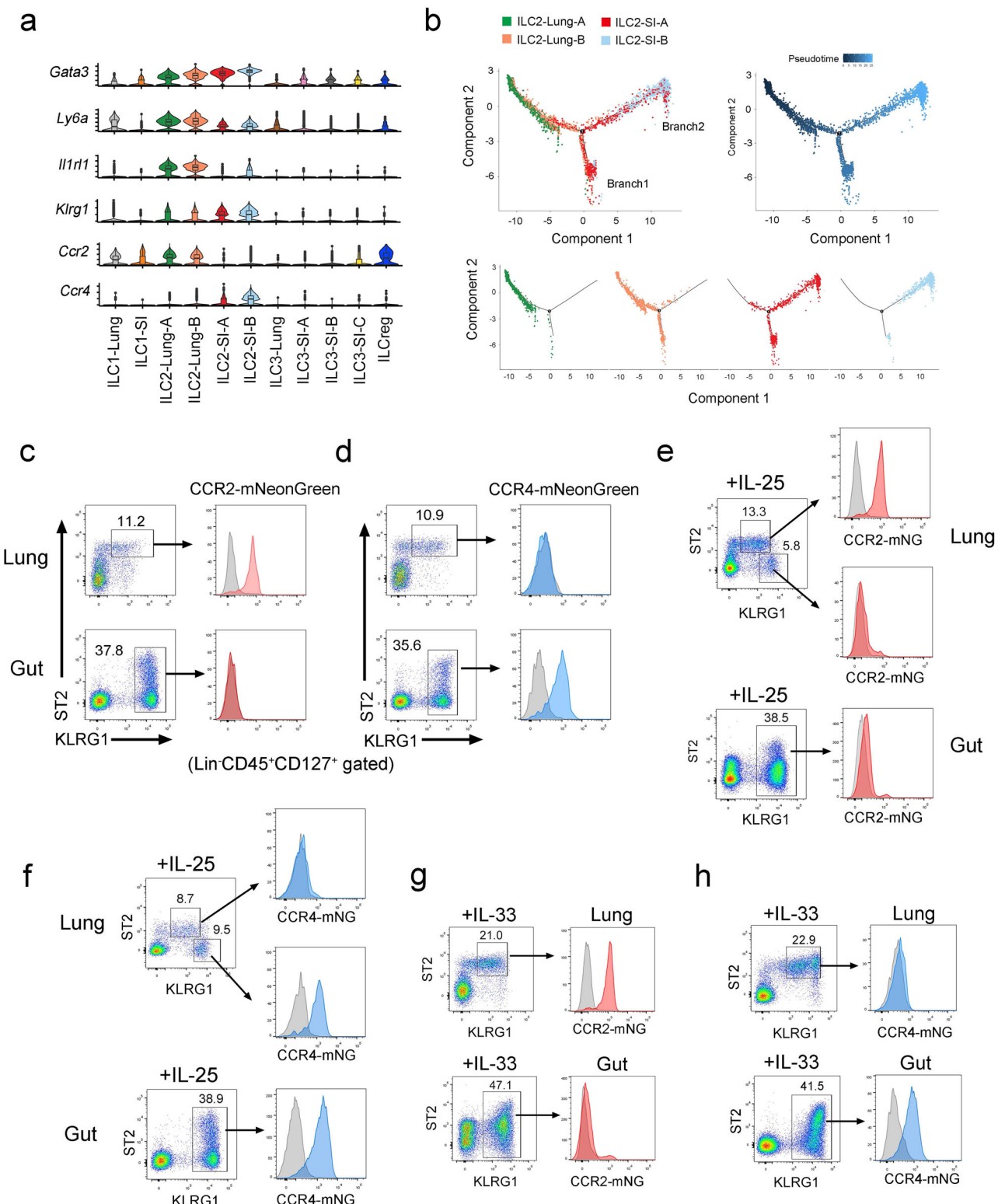

pulmonary ILCs (Fig. 4e and Supplementary Fig. 3j)[23,24]. With the treatment of IL-33, TdTomato-traced lung ILC2s were able to be found in the gut, indicating the translocation of gut ILC2s from lung ILC2s (Fig. 4e and Supplementary Fig. 3k). To further investigate the maturation process of ILC2s in vivo, we generated an ILC2-depleted mouse model by using a diphtheria toxin (DT)-induced cell ablation system. Through the use of a lineage tracing system, we noticed that CCR2 was expressed in lung ILC2s and that gut ILC2s used to be CCR2+ (Supplementary Fig. 4a). Thus, *Ccr2-mNeonGreen-Cre; Rosa26-STOP-*

*DTR* mice that expressed DT receptor on ILC2s can be used for the depletion of both lung and gut ILC2s (Supplementary Fig. 4b). With the treatment of DT, ILC2s in the lung and gut were eliminated. ILC2Ps in BM and ILC3s were not impaired (Fig. 4f, g; Supplementary Fig. 4c). However, the depletion of CCR2+ monocytes did not affect the ILC2 population (Supplementary Fig. 4d, e). Next, we used IL-33 to promote the development and maturation of ILC2Ps from BM for the reconstruction of lung and gut ILC2s. Consistent with the adoptive transfer data, IL-33 administration induced the reconstruction of ILC2s first in

**Fig. 3 | CCR2 and CCR4 serve as tissue specific markers of ILC2s. a** Violin plots showed ILC2 signature genes in the lung and the intestine. *Ccr2* was highly expressed in lung ILC2s and *Ccr4* is restricted to intestine ILC2s. Box plots indicate median (middle line), 25th, 75th percentile (box) of the data and the maximum and minimum values as endpoints for the whiskers. **b** Maturation trajectory of lung and intestine ILC2s were analyzed by Monocle pseudotime analysis. **c, d** Expression of CCR2 and CCR4 on ILC2s. CCR2-mNeonGreen reporter mice (**c**) and CCR4-mNeonGreen reporter mice (**d**) were generated as described in "Methods" and analyzed for the expression of CCR2 and CCR4 on ILC2s by flow cytometry. Lin⁻CD45⁺CD127⁺ lymphocytes were gated for analysis of KLRG1 and ST2. (Lin=CD3e,CD8a,CD19,CD11b,CD11c,Gr1,F4/80,Ter119). **e, f** CCR2-mNeonGreen and CCR4-

mNeonGreen mice were intraperitoneally injected with 200 ng/mouse/day IL-25 for three constitutive days. Expression of CCR2-mNeonGreen (**e**) and CCR4-mNeonGreen (**f**) on ILC2s from respective mice were analyzed by flow cytometry. **g, h** CCR2-mNeonGreen mice (**g**) and CCR4-mNeonGreen mice (**h**) were analyzed for the expression of CCR2-mNeonGreen and CCR4-mNeonGreen on ILC2s after i.p. injection of 400 ng/mouse/day IL-33 for three constitutive days. (Gate strategies for ILC2s are: Lin⁻CD127⁺ST2⁺KLRG1⁺ for lung ILC2s, Lin⁻CD127⁺ST2⁻KLRG1⁺ for iILC2, Lin⁻CD127⁺KLRG1⁺ for gut ILC2s, Lin=CD3e,CD19,CD11b,CD11c,Gr1,F4/80,NK1.1). Data are representative at least three independent experiments. Source data are provided as a Source Data file.

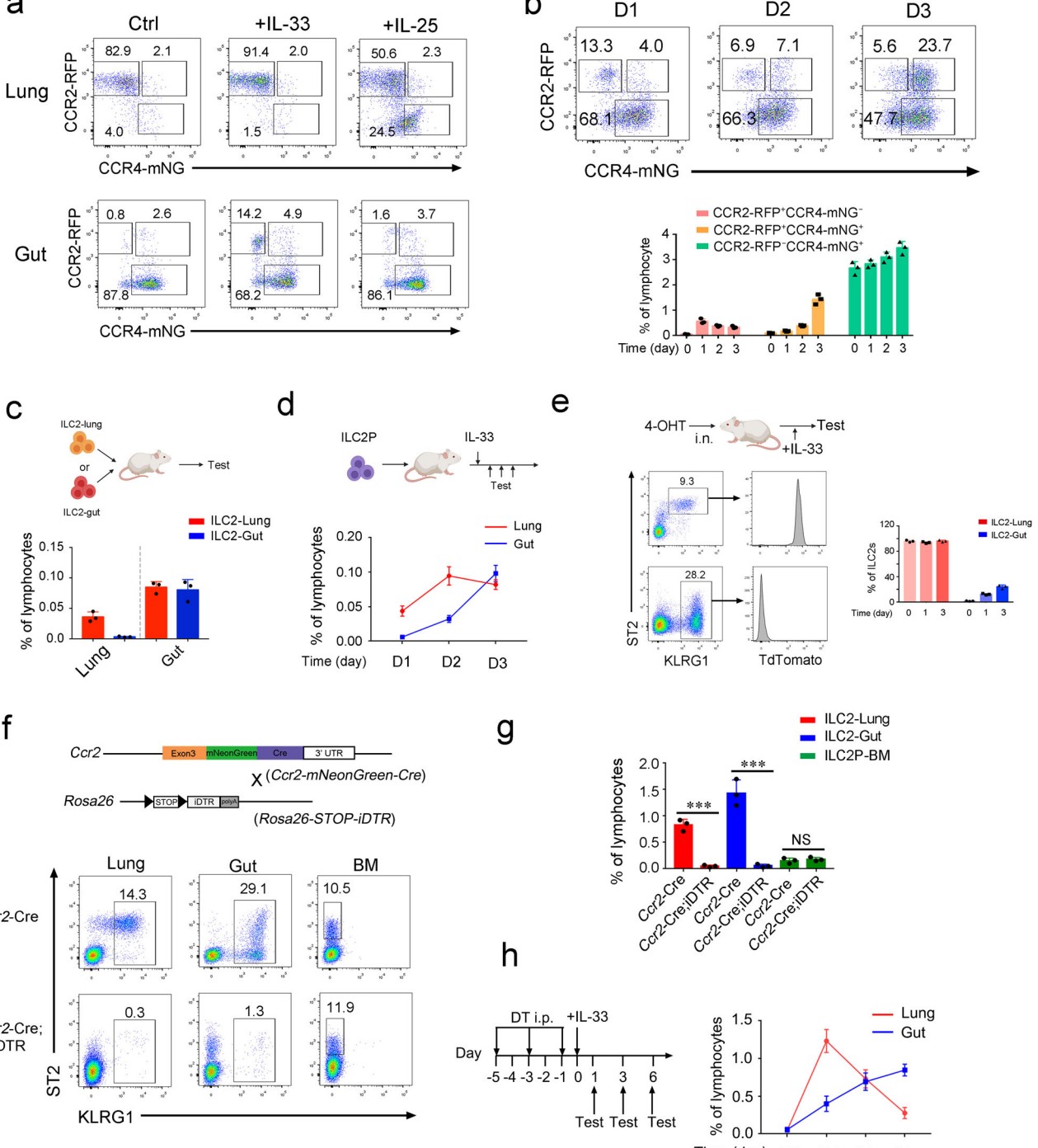

**Fig. 4 | Maturation of ILC2s through lung-gut axis. a** CCR2-RFP;CCR4-mNeon-Green mice were i.p. injected with IL-25 or IL-33. After 24 h, expression of mNeonGreen and red fluorescent protein (RFP) in lung or gut ILC2s was analyzed by flow cytometry. **b** Analysis of mNeonGreen and RPF expression of gut ILC2s from CCR2-RFP;CCR4-mNeonGreen mice after intraperitoneal administration of IL-33 for the indicated days. The cell frequency of indicated subgroups of gut ILC2s was calculated and shown as mean ± SD (lower panel). (*n* = 3 for each group). **c** Adoptive transfer of lung or gut ILC2s (5 × 10⁴) from WT CD45.1 mice to NOD-*Prkdc^scid^IL2rg^tm1^*/Bcgen (B-NDG) mice by intravenous injection. Six days after of transfer, lung and gut ILC2s from the recipient mice were analyzed by flow cytometry. The cell frequency of ILC2s were calculated and shown as mean ± SD. (*n* = 3 for each group). **d** Adoptive transfer and development of ILC2Ps in the lung and gut. ILC2Ps (Lin⁻CD127⁺ST2⁺Sca1⁺KLRG1⁻) from BM of CD45.1 mice were isolated and intravenously injected into B-NDG mice. One week after transfer, the recipient mice were i.p. injected with 400 ng/mouse/day IL-33 for the indicated days. The cell frequency of ILC2s in the lung and gut were calculated and shown as mean ± SD. (*n* = 3 for each group). **e** Tracing of lung ILC2s. Solubilizing 4-OHT (2 µg/g mouse) were atomized and delivered into lung of *Id2-Cre/ERT2;Rosa26-STOP-tdTomato* mice by liquid aerosol devices as described in "Methods". Expression of TdTomato in ILC2s were

analyzed by flow cytometry two days after 4-OHT treatment (left panel). Two days after 4-OHT treatment, *Id2-Cre/ERT2;Rosa26-STOP-tdTomato* mice were i.p. injected with IL-33. The frequency of TdTomato⁺ ILC2s from the mice treated IL-33 with indicated days were calculated and shown as mean ± SD. The schematic diagrams in (**c–e**) were created with BioRender.com. **f, g** Depletion of ILC2s in the lung and gut. *Ccr2-mNeonGreen-Cre;Rosa26-STOP-DTR* mice were subjected to i.p. injection of 100 ng/mouse diphtheria (DT) every two days for six days. Cell frequency of ILC2s from the lung, gut and BM were analyzed by flow cytometry (**f**) and shown as mean ± SD (**g**). (*n* = 3 for each group). ***, *P* < 0.001 by Two-tailed unpaired Student's *t*-test. NS, not significant (*P* > 0.05) by Two-tailed unpaired Student's *t*-test. (*P* = 0.0003, 0.0007, 0.4971 for Lung, Gut, BM respectively by Two-tailed unpaired Student's *t*-test). **h** The reconstitution of ILC2s in the lung and gut after ILC2 depletion. ILC2s were depleted as shown in (**f**) and the ILC2-depleted mice were subjected to the intraperitoneal injection of 400 ng/mouse IL-33 at day 0. After the indicated days, cell frequency of lung and gut ILC2s were analyzed by flow cytometry and shown as mean ± SD. (*n* = 3 for each group) Data are representative of at least three independent experiments. Source data are provided as a Source Data file.

the lung followed by the gut (Fig. 4h). Together, these results indicate that ILC2s undergo translocation through the lung-gut axis.

## CCR2 and CCR4 of ILC2s serve different functions in the lung and gut

With the expression of CCR2 in the lung and CCR4 in the gut, ILC2s may harbor differential functions during stimulation. To ascertain if CCR2 or CCR4 was required for the maintenance of ILC2s in their local tissues, we obtained CCR2-deficient mice and analyzed lung and gut ILC2 levels under both steady state and stimulatory conditions. CCR2 deficiency did not impact the ILC2 count at steady state (Fig. 5a). However, upon stimulation by IL-25 or IL-33, ILC2 counts in both the lung and gut were substantially decreased (Fig. 5a, b). Notably, the ligands of CCR2 and CCR4 were induced upon IL-25 or IL-33 stimulation (Supplementary Fig. 5a). We next evaluated the level of ILC2Ps/ILC2s in the BM. We found that ILC2Ps/ILC2s were accumulated in the BM of CCR2 deficient mice upon stimulation (Fig. 5c) indicating the requirement of CCR2 for ILC2 relocalization from BM to the lung. CCR2-deficient ILC2s might not egress from BM and settle in the lung under cytokine stimulation. CCR2 was not expressed in gut ILC2s, and *Ccr2* expression did not induced in gut ILC2s upon IL-33 treatment (Supplementary Fig. 5b). Due to the loss of lung ILC2s, gut ILC2s might not be properly supplemented—this would explain the decreased level of ILC2s in the gut of *Ccr2^-/-^* mice. Next, we transplanted WT and *Ccr2^-/-^* ILC2Ps into the immunodeficiency mice for ILC2 reconstitution (Fig. 5d). With the treatment of IL-33, ILC2 count in BM was apparently increased and failed to be reconstituted in the lung and gut (Fig. 5d). Thus, these data elucidated that *Ccr2^-/-^*ILC2s failed to egress to the lung, and gut ILC2s were affected by the defect of lung ILC2s.

CCR2 deficiency not only impacted the count of ILC2s in the lung and gut, but also led to the dysfunction of ILC2s in these two mucosal sites. ILC2s found in the lung and gut of CCR2 deficient mice tended to be more inflammatory. *CCR2^-/-^* ILC2s highly expressed *Il17a*, *Il17f*, *Icos*, and *Agr1* which serve as activation markers for inflammatory ILC2s (Fig. 5e)[25,26]. Thus, CCR2 deficiency affects both the location and function of ILC2s in the lung and gut.

To investigate the activity of CCR4 in ILC2s, we next analyzed the function of ILC2s in CCR4-deficient mice (Fig. 5f). There were no significant changes in gut ILC2 count between control and *CCR4^-/-^* mice under steady state and stimulatory conditions. However, the presence of inducible iILC2s in the lung was increased upon IL-25 stimulation. We therefore proposed that CCR4 is required for the maintenance of ILC2s in the gut. Next, we adoptively transferred WT and *Ccr4^-/-^* ILC2Ps for ILC2 reconstitution and found that *Ccr4* deficiency did not affect the ILC2 population under steady state or stimulatory condition with IL-33. However, the iILC2s were augmented upon IL-25 treatment

(Supplementary Fig. 5c). Additionally, transcriptome analysis showed that *CCR4^-/-^* iILC2s expressed high levels of *S1pr4* and *Cd37*, which are related to the cell trafficking (Fig. 5g)[27,28]. CCR4 deficiency promoted the translocation of iILC2s to the lung, supporting the idea that CCR4 is required for the residence of iILC2s in the gut upon stimulation. In sum, CCR2 and CCR4 exert unique functions for lung and gut ILC2s under various stimulation scenarios.

## Postnatal development of lung ILC2s affects the function of gut ILC2s

During the postnatal phase, it was reported that lung ILC2s accumulated at postnatal day (PND) 10–14[29,30]. Postnatal hyperactivity of type II immune response apparently promoted the development of asthma and airway hyperresponsiveness (AHR)[31]. Consistently with previous study, we found that expansion of lung ILC2 appeared earlier compared to gut ILC2s (Fig. 6a, b) at postnatal stage. By using CCR2-RFP;CCR4-mNeonGreen reporter mice, we found that the transition of RFP⁺ILC2s to mNG⁺ILC2s in the gut of postnatal mice (Fig. 6c and Supplementary Fig. 5d). Next, we transferred lung or gut ILC2s from postnatal mice into immunodeficient mice. Notably, lung ILC2s from postnatal mice were able to translocate to the gut and generated gut ILC2s, but not the reverse (Fig. 6d), suggesting that the lung-gut maturation axis may also occur during postnatal development of ILC2s.

During the development of lung, allergens such as house dust mite (HDM) is considered as a major risk factor of asthma[32]. Asthma and IL-33 signaling are also related to some cases of the intestinal inflammation[33,34]. Therefore, we questioned whether the dysfunction of lung ILC2s during postnatal stage also affected the homeostasis of gut ILC2s. We treated postnatal mice with HDM, and increased numbers of ILC2s were observed both in the lung and gut of HDM-treated mice (Fig. 6e). After two weeks, we challenged HDM-treated mice with IL-33 to investigate the long-lasting effects of ILC2s. We found that HDM-treated mice had higher numbers of ILC2 compared with PBS controls (Fig. 6e). Transcriptome data showed that HDM treatment increased the levels of inflammatory cytokines and molecules in gut ILC2s, including *Il4*, *Il5*, *Il6*, *Il13*, *Nmur1*, *Arg1*, and so on (Fig. 6f). The production of allergenic cytokines was also increased (Fig. 6g), suggesting the dysfunction of ILC2s in the gut. In sum, allergic airway inflammation that induces the dysfunction of lung ILC2s also aroused the hyperreaction of gut ILC2s along lung-gut axis.

In sum, our data imply that lung ILC2s tend to express CCR2, whereas intestinal ILC2s express CCR4. ILC2s undergo maturation through the lung-gut axis after egress from BM. During the development of ILC2s at postnatal stage, trigger of allergic asthma in the lung induced dysfunction of gut ILC2s.

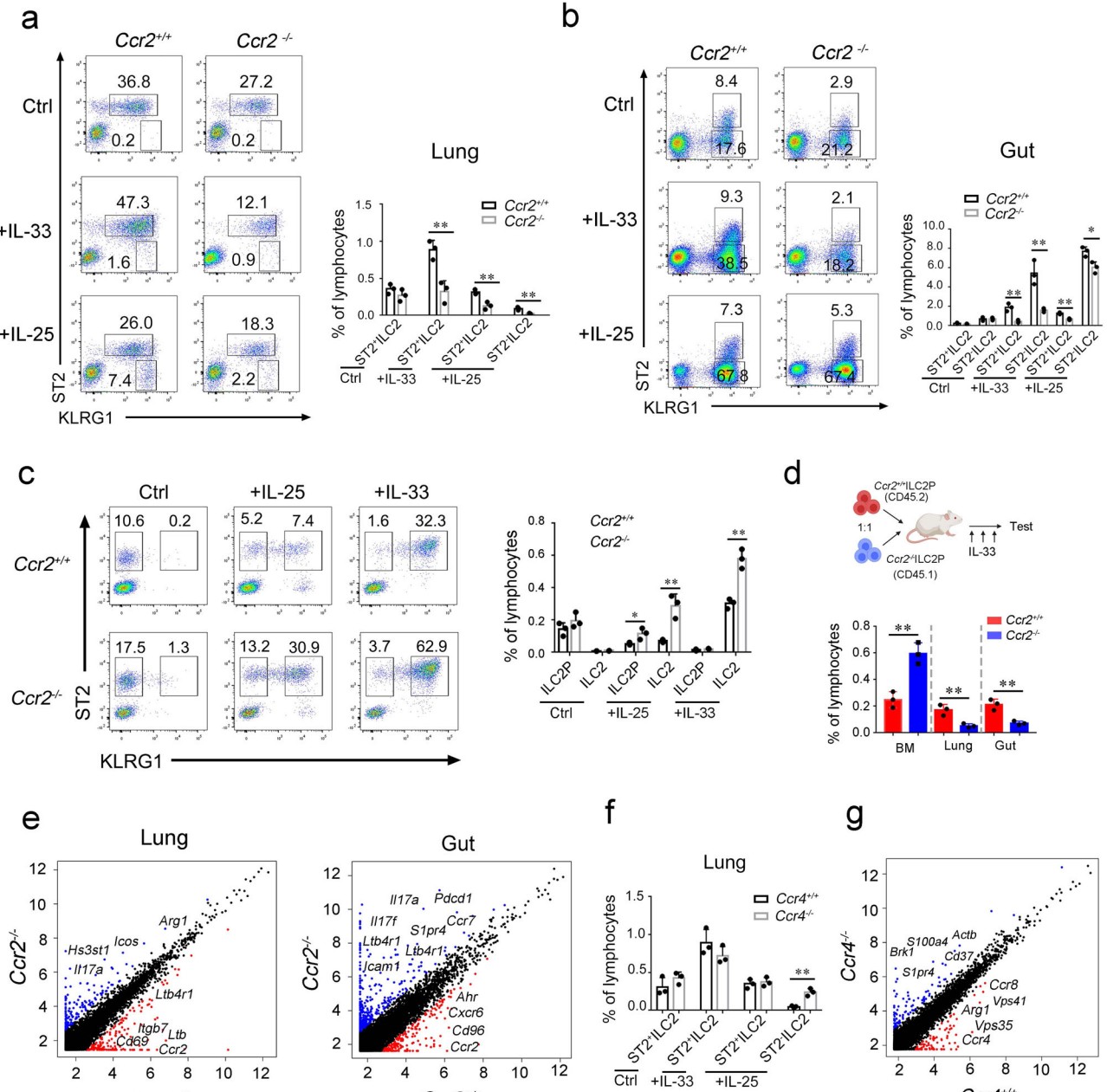

**Fig. 5 | The differential function of CCR2 and CCR4 in lung and gut ILC2s.**
**a**, **b** Deficiency of CCR2 abrogated the ILC2 counts in the lung (**a**) and gut (**b**). WT or *Ccr2*−/− mice were subjected to the i.p. injection of IL-25 or IL-33 followed by flow cytometry analysis. The cell frequency of indicated ILC2 subsets was calculated and shown as mean ± SD (right panel). (*n* = 3 for each group) (*P* = 0.0072, 0.0059, 0.0056 for **a**, *P* = 0.0040, 0.0066, 0.0012, 0.0196 for **b**). **c** CCR2 is required for the ILC2 relocalization from BM to the lung. ILC2Ps (Lin−CD127+Sca1+ST2+KLRG1−) and ILC2s (Lin−CD127+Sca1+ST2+KLRG1+) from BM of WT or *Ccr2*−/− mice were analyzed by flow cytometry. The indicated ILC2 subsets was calculated and shown as mean ± SD (right panel). (*n* = 3 for each group) (*P* = 0.0376, 0.0064, 0.0021). **d** Adoptive transfer of WT and *Ccr2*−/− ILC2Ps. ILC2Ps (Lin−CD127+Sca1+ST2+KLRG1−) were isolated from BM of WT and *Ccr2*−/− mice. WT and *Ccr2*−/− ILC2Ps were 1:1 mixed (5×10⁴ for each) and i.v. injected into B-NDG mice. One week after transfer, the recipient mice were i.p. injected with 400 ng/mouse/day IL-33 for three days. The cell frequency of ILC2s in the lung and gut were calculated and shown as mean ± SD. (*n* = 3 for each group). (*P* = 0.0039, 0.0086, 0.0050 by Two-tailed unpaired

Student's *t*-test). The schematic diagram was created with BioRender.com. **e** Scatter plot comparing the gene expression pattern between WT ILC2s versus *Ccr2*−/− ILC2s from the lung and gut. ILC2s (Lin−CD45+CD127+KLRG1+) were isolated from WT or *Ccr2*−/− mice after treatment with IL-33 and subjected to bulk mRNA sequencing. Blue dots, upregulated genes in *Ccr2*−/− ILC2s; red dots, upregulated genes in WT ILC2s. Representative differentially expressed genes (DEGs) are depicted. **f** WT or *Ccr4*−/− mice were i.p. injected with IL-25 or IL-33 for three days. ILC2s from the lung and gut were analyzed by flow cytometry and cell frequency was shown as mean ± SD. (*n* = 3 for each group) (*P* = 0.0031 by Two-tailed unpaired Student's *t*-test). **g** Comparison of gene expression in lung iILC2s from WT versus *Ccr4*−/− mice. iILC2s (Lin−CD45+CD127+ST2−KLRG1hi) were isolated from the lung of WT or *Ccr4*−/− mice after treatment with IL-25 followed by bulk mRNA sequencing. Blue dots, upregulated genes in *Ccr4*−/− ILC2s; red dots, upregulated genes in WT ILC2s. Representative DEGs are depicted. Data are representative of at least three independent experiments. *, *P* < 0.05; **, *P* < 0.01 by Two-tailed unpaired Student's *t*-test. Source data are provided as a Source Data file.

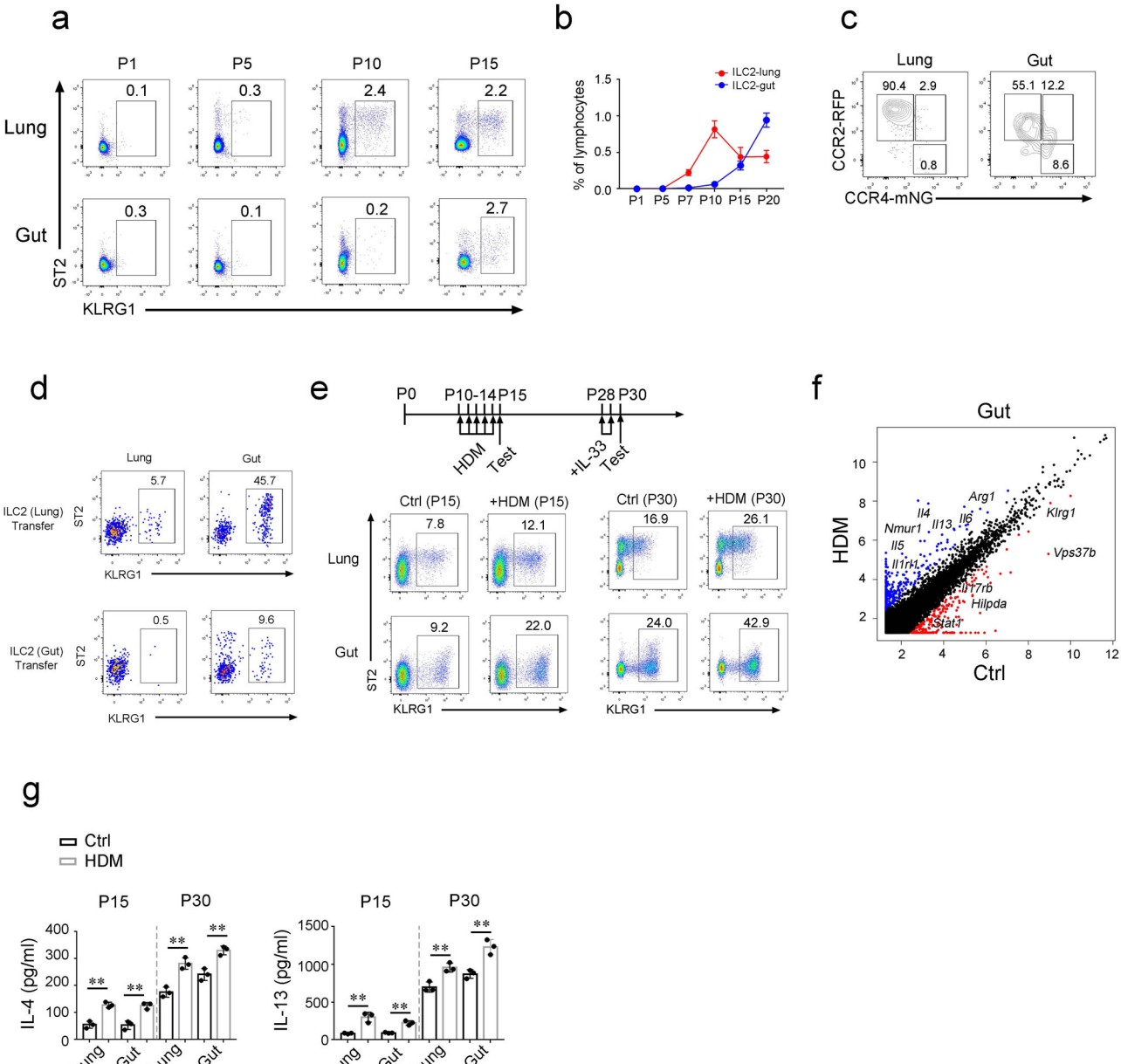

**Fig. 6 | Postnatal development of lung ILC2s contributes to the proper function of gut ILC2s. a, b** Analysis of lung and gut ILC2s from mice of indicated postnatal (P) days by flow cytometry and show as mean ± SD. **c** Expression of CCR2 and CCR4 on ILC2s from CCR2-RFP;CCR4-mNeonGreen mice at postnatal day 12 was analyzed by flow cytometry. **d** Transfer of lung ILC2s or gut ILC2s isolated from mice at postnatal day 12 to the B-NDG mice. Seven days after transfer, ILC2s from the lung and gut of B-NDG mice were analyzed by flow cytometry. **e** Treatment of HDM at early age affected the function of lung and gut ILC2s. Mice at postnatal day (P) 10 were i.n. treated with 10 μg HDM for 5 consecutive days, and lung or gut ILC2s were subsequently analyzed by flow cytometry (indicated as P15). Two weeks after first round of HDM treatment, mice were i.p. challenged with 400 ng/mouse/

day IL-33 for two days and subjected to flow cytometry analysis (indicated as P30). **f** Postnatal mice were treated as (**e**), gut ILC2s from mice with IL-33 treatment at P30 were analyzed by bulk mRNA-sequencing. Blue dots, upregulated genes in HDM-treated (HDM) mice; red dots, upregulated genes in PBS-treated (Ctrl) mice. Representative DEGs are depicted. **g** Secretion cytokines of ILC2s from HDM-IL-33 treated mice or PBS treated mice (Ctrl) in (**e**) were analyzed by ELISA and show as mean ± SD. Data are representative of at least three independent experiments. **, $P < 0.01$ by Two-tailed unpaired Student's $t$-test. ($P = 0.0015$, 0.0030, 0.0029, 0.0048 for IL-4 group; $P = 0.0048$, 0.0017, 0.0076, 0.0056 for IL-13 group). Source data are provided as a Source Data file.

## Discussion

Innate lymphoid cells are considered tissue-resident lymphocytes and serve protective and regulatory functions in mucosal sites. The functions of ILCs in mucosal tissues have been well-characterized in previous studies[35]. However, the maturation processes of ILCs in various mucosal sites are largely unknown. In this study, we identified the subsets and functions of ILCs in the lung and gut and found that they were significantly different from one another. Moreover, lung ILC2s were characterized by high levels of CCR2 while CCR4 was specific to

gut ILC2s. The pathologic allergens that affected the function of lung ILC2s also gave rise to inflammatory ILC2s in gut. Thus, our data show that both function and maturation process are related between a variety of mucosal tissues.

The development of ILCs is a well-organized process that requires specific factors. ILCs are abundant in mucosal sites, which are commonly exposed to external antigens. According to our data, the differences in ILCs between tissues is much larger than between ILC subsets. ILCs at particular mucosal sites likely adapt to their

microenvironment and maintain an imprint of their location[35]. Extrinsic factors from mucosal niches might contribute to the establishment of tissue-specific features. Microbiota and metabolites in certain mucosal sites could possibly be factors that determine the characteristics of ILCs in certain tissues. For example, we found that gut ILCs upregulate the molecules involved in metabolism. Gut ILC2s express high levels of *Hilpda*, a protein coding gene involved in lipoprotein metabolism and lipid storage. This would indicate that gut ILC2s may take part in the regulation of metabolite levels. Gut ILC3s display high expression level of metabolite-sensing receptor *Ffar2*, which promotes ILC3 expansion and function[19]. Overall, the metabolites in the gut might change the features and functions of ILCs, and ILCs may also take part in the storage and consumption of metabolites. The relationships between ILCs and metabolites are worthy of investigation in future studies.

The mucosal immune system displays many similarities and connections throughout a variety of mucosal sites. The concept of the common mucosal immune system (CMIS) was introduced to illustrate the similar immune responses at mucosal sites that do not appear in systemic immune system[36]. Although effector function is correlated, the maturation of immune cells at various mucosal sites was thought to be separate. Our data provides evidence for a previously unrecognized relationship in ILC maturation between the lung and the gut. We found that the maturation trajectory of ILC2s occurs through the lung-gut axis. During the ontology of ILC2s under stimulation or in the postnatal stage, lung ILC2s appear earlier compared to gut ILC2s. The intermedium stage of ILC2s (CCR2+CCR4+) could be found in the gut but not in the lung, suggesting that these cells transition from lung ILC2s to gut ILC2s. In other words, the transfer of lung ILC2s gives rise to gut ILC2s. The question of whether there are other immune cells that follow this rule requires further research. The maturation order of ILC2s at mucosal sites raised an important question: Is it absolutely necessary for the maturation of ILC2s? To answer these questions, we took advantage of knockout mouse models for critical markers of ILC2s in mucosal sites. A deficiency of CCR2 resulted in the augmentation of ILC2Ps in BM and the reduction of ILC2s in the lung. The defect in transfer of ILC2s to the lung also abrogated the count of gut ILC2s, even though gut ILC2s do not express CCR2. Thus, the maturation process of lung ILC2s is required for the homeostasis of gut ILC2s.

Treatment with IL-25 or IL-33 indeed shows different regulation mechanism. IL-25 treatment induces the migration of ILC2 from gut to the lung[37]. However, several reports show the IL-33 induced-migration of ILC2s. Locksley's group reported that during infection, second wave of ILC2s induced by IL-33 were from the lung and was abrogated in IL-33R−deficient mice[38], indicating that IL-33 might induce the ILC2 migration from the lung. Moreover, IL-33 promotes the egress of ILC2s from BM, and critically enhances ILC2 hematogenous migration and tissue repopulation in a lung-tropic manner, indicating that IL-33 might induce different regulation pathway from IL-25.

During the postnatal phase, ILC2s were accumulated in the lung and then appeared in the gut[29]. We also observed the lung-gut maturation of ILC2s in the postnatal stage. Pathologic allergens such as house dust mite (HDM) that promotes asthma increased the numbers of lung ILC2s, as well as gut ILC2s. Accumulating evidences showed that asthma and IL-33 signaling were related to intestinal inflammation. Clinic studies demonstrated that asthma is associated with subsequent development of colitis, especially ulcerative colitis among individuals under 16-year-old[33], indicating that hyperreaction or dysfunction of pulmonary immune system at early age might affect the homeostasis of the intestine. Whether human ILC2s undergo development along lung-gut axis and obtained appropriate activities are worthy to be further investigated. Recently, reports showed that tissue-resident ILC progenitors were found in the lung that gave rise to ILC2 population[20,39]. They also found that BM was a source of full spectrum

of ILC2s during infection. It is possible that different kinds of stimulation might result in various ILC2 differentiation trajectories

In summary, our study profiled ILC subsets in the lung and the intestine and identified their tissue-specific markers. Our findings imply that ILC2s undergo maturation through the lung-gut axis to obtain proper function. A defect of ILC2 development in the lung significantly impacts the count and function of ILC2s in the gut. Therefore, our study provides insight into the relationship in ILC maturation between various mucosal sites with potential implications for inflammation, allergy, and causes of immune disorder during the establishment of the mucosal immune system.

## Methods
### Study approval
All animal procedures were approved by the Institutional Ethics Committee of Institute of Microbiology, Chinese Academy of Sciences. The study is compliant with all relevant ethical regulations regarding animal research.

### Antibodies and reagents
Antibodies used for flow cytometry are as follows: anti-mouse CD3-eFluor 450 (Invitrogen, Cat# 48-0032-82, 1:500), anti-mouse CD19-eFluor 450 (Invitrogen, Cat# 48-0193-82, 1:500), anti-mouse CD8a-eFluor 450 (Invitrogen, Cat# 48-0081-82, 1:500), anti-mouse CD11b-eFluor 450 (Invitrogen, Cat# 48-0112-82, 1:500), anti-mouse CD11c-eFluor 450 (Invitrogen, Cat# 48-0114-82, 1:500), anti-mouse Gr1-eFluor 450 (Invitrogen, Cat# 48-5931-82, 1:1000), anti-mouse F4/80-eFluor 450 (Invitrogen, Cat# 48-4801-82, 1:500), anti-mouse Ter119-eFluor 450 (Invitrogen, Cat# 48-5921-82, 1:1000), anti-mouse CD45.2-PE/Cyanine7 (Biolegend, Cat# 109829, 1:1000), anti-mouse CD127-PerCP-eFluor 710 (Invitrogen, Cat# 46-1273-82, 1:300), anti-mouse CD117-PE/Cyanine7 (Invitrogen, Cat# 25-1171-82, 1:500), anti-mouse NK1.1-eFluor 450 (Invitrogen, Cat#48-5941-82, 1:500), anti-mouse ST2-PE (Invitrogen, Cat# 12-9333-82, 1:500), anti-mouse KLRG1-APC (Invitrogen, Cat# 17-5893-82, 1:500), anti-mouse RORγt-APC (Invitrogen, Cat# 17-6988-82, 1:300), anti-mouse CD45.1-PE/Cyanine7 (Biolegend, Cat# 110715, 1:500), anti-mouse CCR2-PE/Cyanine7 (Biolegend, Cat# 150611, 1:100), anti-CCR4-PE/Cyanine7 (Biolegend, Cat# 131213, 1:100), and anti-mouse Sca1-FITC (Biolegend, Cat#108105, 1:500). Recombinant mouse IL-25 (R&D System, Cat# 1399-IL-025) and mouse IL-33 (R&D System, Cat# 3626-ML-010) were purchased from R&D system. 7-AAD (Cat# 00-6993-50, 1:1000) was from Invitrogen.

### Mice
*Ccr2-mNeonGreen-Cre*, *Cc4-mNeonGreen-Cre*, and *Ccr4−/−* mice were generated by Cyagen Biosciences. For *Ccr2-mNeonGreen-Cre* mice and *Cc4-mNeonGreen-Cre* mice, TAA stop codon of *Ccr2* or *Ccr4* gene was replaced with 'P2A-mNeonGreen-T2A-Cre' cassette by using CRISPR/Cas9 technology. The correct gene targeting was confirmed by Southern blot. F1 mice were generated by breeding of F0 mice with WT mice. Targeted alleles were confirmed by PCR screening and DNA sequencing. The primers for genotyping are: *Ccr2-mNeonGreen-Cre*-F1: 5'-ACAGAGACTCTTGGAATGACACACT-3', *Ccr2-mNeonGreen-Cre*-R1: 5'-ATAACGTTCTGGGCACCTGATTTA-3', *Ccr2-mNeonGreen-Cre*-R2: 5'-CTTGCTTTAACAGAGAGAAGTTCGTG-3', *Ccr4-mNeonGreen-Cre*-F1: 5'-AGTTTCAATACCGGAGATCATGC-3', *Ccr4-mNeonGreen-Cre*-R1: 5'-CTTGCCATGGTCTTGGTTTTACT-3',*Ccr4-mNeonGreen-Cre*-R2: 5'-GAGAGGTACCTAGACTACGCCAT-3'. For *Ccr4−/−* mice, exon 2 of *Ccr4* gene was deleted by using CRISPR/Cas9 technology. Targeted alleles were confirmed by PCR screening and DNA sequencing. The primers for genotyping are: *Ccr4*-F1: 5'-CTCAGGCATAGTGACAGGTATCC-3', *Ccr4*-R1:5'-ACCATGCCCAATACCCAATAACAG-3', *Ccr4*-R2:5'-ATAGATGGGGATAGAAACCCCGAA-3'.

*Ccr2^RFP/RFP^* and Rosa26-STOP-DTR mice were from Jackson laboratory. *Rosa26-STOP-tdTomato* mice were from Shanghai Research

Center for Model Organisms. B-NDG (NOD-*Prkdc^scid^IL2rg^tm1^*/Bcgen) mice were from Beijing Biocytogen. ILC2 depletion mice *Rosa26-STOP-DTR;Ccr2-mNeonGreen-Cre* were obtained by crossing *Rosa26-STOP-DTR* mice with *Ccr2-mNeonGreen-Cre* mice. Both female and male mice were used in experiments. Age- and sex-matched littermates between 8 and 16 weeks of age were used. Mice were assigned randomly to experimental groups.

CD45.1 and B-NDG mice were BALB/c background. All the other mouse strains are C57BL/6 background. Both male and female mice were used. Mice were maintained under specific pathogen-free conditions.

## Isolation of ILCs from lung and intestine

For lung ILCs: Lung tissues from mice were cut into pieces and placed in RPMI-1640 containing 2% (vol/vol) FBS (Thermo Fisher Scientific), Collagenase II and III (1 mg/ml; Worthington), DNase I (200 µg/ml; Roche), and incubated for 45 min at 37 °C. For intestinal ILCs: Intestines of mice were isolated and cleaned. The intestines were cut into pieces after the removing of Peyer's patches. Epithelial layers were removed by incubation three times in 5 mM EDTA Ca$^{2+}$ and Mg$^{2+}$ free Hank's medium for 20 min each at 37 °C, and the epithelial cells were collected if needed. Then, intestines were cut into fine pieces and digested twice for 45 min each at 37 °C with RPMI-1640 containing 2% (vol/vol) FBS, Collagenase II and III (1 mg/ml; Worthington), DNase I (200 µg/ml; Roche) and dispase (4U/ml; Sigma). All cell suspensions were passed 70 µm cell strainers, and washed twice with PBS to recover cells.

Lin$^-$CD45$^+$ cells were sorted out by Magnetic Cell Sorting system (MACS, Miltenyi Biotec). Sorted cells were blocked with anti-CD16/32 antibody for 30 min on ice and then stained with anti-CD45, anti-CD127, and lineage cocktail antibodies for total ILC population or stained with anti-KLRG1, anti-ST2, anti-CD127, anti-CD45, and lineage cocktail for ILC2s on ice for 1 h. After staining with 7AAD, cells were isolated by flow cytometer (FACS Aria III, BD), and ILCs (Lin$^-$CD45$^+$CD127$^+$7AAD$^-$) or ILC2s (Lin$^-$CD45$^+$CD127$^+$KLRG1$^+$7AAD$^-$) were isolated and subjected to single cell RNA sequencing or adoptive transfer. Purity of ILCs was over 95% for each assay that was determined by post sorting analysis of flow cytometry. (lineage= CD3, CD4, CD8, CD19, CD11b, CD11c, Gr1, F4/80, and Ter119 for total ILCs; lineage= CD3, CD4, CD8, CD19, CD11b, CD11c, Gr1, F4/80, Ter119 and NK1.1 for ILC2s).

## scRNA-seq cDNA library preparation and sequencing

ILCs isolated from lung or small intestine were harvested and sorted by flow cytometry (*n* = 5 for each group). The mice share the similar distribution of ILCs from the same tissue and we pooled them together as one sample for scRNA-seq. Two biological repetitions were made for each sample. ILCs (Lin$^-$CD45$^+$CD127$^+$) (Lin=CD3e,CD8a,CD19,CD11b, CD11c,Gr1,F4/80,Ter119) were loaded on a Chromium Single Cell Controller (10×Genomics) to generate single-cell Gel Bead-In-Emulsions (GEM). cDNA libraries were prepared by using Single Cell 3′ Library and Gel Bead Kit V3 (10×Genomics) according to the manufacturer's protocol followed by sequencing on an Illumina HiSeq X Ten (PE150).

## Bulk mRNA sequencing

ILC2s (Lin$^-$CD45$^+$CD127$^+$KLRG1$^+$) were sorted and then resuspended in Trizol (Life Technologies). RNA was subsequently extracted with phenol-chloroform method. Then amplification was carried out using the Smart-Seq2 method. An Oligo-dT primer was introduced to the reverse transcription reaction for first-strand cDNA synthesis, followed by PCR amplification and purification steps. Then the cDNA was sheared randomly by ultrasonic waves for Illumina library preparation protocol including DNA fragmentation, end repair, 3′ ends A-tailing, adapter ligation, PCR amplification and library validation. Qualified libraries were then loaded on Illumina Hiseq X Ten platform (PE150) for sequencing.

## scRNA-seq data preprocess

We processed the single cell RNA sequencing data according to previous study[40]. We aligned reads to the mouse mm10 genome by using CellRanger software (version 4.0.0) to generate the single cell information. We removed doublets and poor-quality cells by Seurat (version 3.1.5) based on the number of unique molecular identifiers (UMIs) and percentage of mitochondrial genes. Briefly, raw cell counts were filtered in Seurat CreateSeuratObject function with the following criteria: one gene was expressed in at least 3 cells and at least 200 genes were detected in one cell. Mitochondrial genes inside one cell was also calculated with a percentage under 20%. Cells were further filtered with the following requirements: gene numbers inside on cell were no more than 8000 and no less than 200. We obtained data from 8511 ILCs for further analysis. Further analyses including normalization, scaling, clustering of cells, and identifying marker genes were performed by using Seurat (version 3.1.5). "FindVariableGenes" function of Seurat was used to get the variable genes with the following parameters: x.low.cutoff 0.05, x.high.cutoff 8 and y.cutoff 0.5. PCA analysis was conducted using the "RunPCA" function of Seurat and PCs 1–12 were chosen for dimension reduction analysis with Seurat function "RunTSNE". For the clustering analysis, the first twelve PCs were used to calculate clusters with a resolution of 0.7 using Seurat function "FindClusters". Differently expressed genes were calculated through Seurat "FindAllMarkers".

Trajectory analysis of ILC2s was performed by using Monocle (version 2.16.0). We analyzed the maturation trajectory of ILC2s from different tissues according to scRNA-sequencing data from previous report[22]. We ordered the cells onto a pseudotime trajectory based on the immature marker genes in ILC2 precursors and mature genes in ILC2s.

## Flow cytometry

Lymphocytes from the lung and intestine were isolated and blocked with anti-CD16/32 antibody for 30 min on ice. Surface markers were then stained for 1 h on ice. For transcription factor staining, cells were harvested for surface marker staining and fixed and permeablized by FoxP3/Transcription Factor Fixation/Permeabilization buffer set (eBioscience), followed by intracellular antigen staining. Cell suspensions were analyzed by flow cytometer (FACS Aria III, BD)

## Depletion and reconstitution of ILC2s

*Ccr2-mNeonGreen-Cre; Rosa26-STOP-DTR* mice were subjected to i.p. injection of 100 ng/mouse diphtheria (DT) every two days for 3 times. After DT treatment, the depletion of ILC2s from the lung and gut were analyzed by flow cytometry. After DT treatment, *Ccr2-mNeonGreen-Cre; Rosa26-STOP-DTR* mice were subjected to the intraperitoneal injection of 400 ng/mouse/day IL-33 for the reconstitution of ILC2s in the lung and intestine. Cell frequency of ILC2s from the lung and gut were analyzed by flow cytometry.

## Adoptive transfer of ILCs

ILC2s (Lin$^-$CD45.2$^+$CD127$^+$KLRG1$^+$) or ILC2Ps (Lin$^-$CD127$^+$Sca1$^+$ST2$^+$ KLRG1$^-$) were isolated from CD45.1 mice with or without treatment of 400 ng/mouse/day IL-33 as needed. $5 \times 10^4$ ILC2s were adoptively transferred into B-NDG mice. For ILC2 transfer, ILC2s from the lung and gut of recipient mice were analyzed by flow cytometry. For ILC2P transfer, recipient mice were i.p. injected with 400 ng/mouse/day IL-33 for 3 days followed by flow cytometry analysis.

## Tracing of pulmonary ILCs

4-Hydroxy-Tamoxifen (4-OHT) was dissolved in ethanol and Cremophor EL solution and diluted with PBS as reported[41]. *Id2-Cre/ERT2;Rosa26-STOP-tdTomato* mice were anaesthetized and solubilizing 4-OHT (2 µg/g mouse) were atomized and delivered into lung of by Liquid Aerosol Devices (Penn-Century MicroSprayer). Cremophor EL/

PBS without 4-OHT served as vehicle control. ILCs from the treated mice were isolated and analyzed by flow cytometry.

## Reporting summary

Further information on research design is available in the Nature Portfolio Reporting Summary linked to this article.

## Data availability

The scRNA sequencing and bulk RNA sequencing data generated in this study have been deposited in the Genome Sequence Archive in National Genomics Data Center, China National Center for Bioinformation/Beijing Institute of Genomics, Chinese Academy of Sciences database under accession code GSA: CRA009030, CRA009031. Data access can be requested through the GSA access committee, but any queries can be directed to Dr Shuo Wang (wangshuo@im.ac.cn). The remaining data are available within the Article, Supplementary Information or from the author on request. Source data are provided with this paper.

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

## Acknowledgements

We thank Tong Zhao for technical support. We thank Drs. George F Gao (Institute of Microbiology, Chinese Academy of Sciences) and Zusen Fan (Institute of Biophysics, Chinese Academy of Sciences) for valuable advices. This work was supported by the Strategic Priority Research Programs of the Chinese Academy of Sciences (XDB29020000), National Key R&D Program of China (2021YFA1300202, 2022YFC2302900, 2019YFA0111800), National Natural Science Foundation of China (92169113, 32000646), CAS Project for Young Scientists in Basic Research (YSBR-010), Key Research Program of Frontier Sciences of Chinese Academy of Sciences (ZDBS-LY-SM025), Beijing Natural Science Foundation (7212067), Open Project Program of CAS Key Laboratory of Pathogen Microbiology and Immunology, and Youth Innovation Promotion Association of CAS to S.W. Cartoons in Figs.-1a, 4c–e, 5d and Supplementary Figs. 3k, 5c were created with BioRender.com.

## Author contributions

M.Z. performed experiments, analyzed data, and wrote the paper; F.S. and D.Y. performed experiments and analyzed data; J.Z., Z.L., and J.M. performed experiments. P.X. generated animal models and analyzed data. S.W. initiated the study and organized, designed and wrote the paper.

## Competing interests

The authors declare no competing interests.
