## [Peer Review File · Nature Communications]

Maturation and specialization of group 2 innate lymphoid cells through the lung-gut axisReviewer #1 ILC (Remarks to the Author):

This manuscript build on the emerging idea that ILC are strongly influence by their tissue environment. Extending on previously published ILC2 datasets, the authors now add both gut and lung ILC data (which nicely captures ILC1/2/3 and ILCreg). Remarkably, tissue-origin has a much stronger influence than ILC-lineage on the transcriptome. Bioinformatics analysis suggests that lung ILC2 serve as precursors for ILC2 in the gut, and that CCR2/CCR4 represent accurate markers for the lung and gut ILC2. Using adoptive transfer, detailed characterization, lineage trace and depletion experiments, the authors provide very compelling data that support a lung-to-gut developmental axis. One weakness is that the authors do not directly investigate the role of this axis in gut pathologies. The data are very interesting and thought-provoking, as other studies have suggested that ILC2 can locally arise from tissue-resident precursors (i.e. Zeis P 2020, Ghaedi M 2020). Also, given the long-term residence of ILC2 at baseline (i.e. Gasteiger G 2015), it remains uncertain how this mechanism contributes to gut ILC2 homeostasis over longer time periods.

Overall, this manuscript is thought provoking and of broad interest. The authors have generated or used elegant transgenic models, which can be used to address some of the outstanding questions (see below).

Figure 1: The authors perform scRNAseq analysis on lung and gut enriched ILCs. They find the expected ILC subtypes.

a. Do the authors identify IL-18R+ ILC precursors in the lung, which are known to give rise to ILC2/1?

Figure 2: The authors perform more detailed analysis of the scRNAseq dataset, and mainly find that lung ILC express high levels of Ccr2 while gut ILC express more Ccr4.

Figure 3: Focussing on ILC2, the authors show by pseudotime analysis lung ILC2 may serve as a precursor for gut ILC2 subsets. The authors then used Ccr2 and Ccr4 reporter mice to validate their 10X dataset by flow cytometry analysis.

a. The gating strategy for ILC2 is not clear, it appears that the majority of lung ILC2 do not express ST2 or KLRG1, which is i

b. The authors propose that gut ILC2 comprise two rather distinct subgroups, which arise from a common precursor. What are the defining functional genes of the two gut subgroups (at their most mature end-states), and could the authors speculate about their function in homeostatic conditions?

c. While the authors state that lung ILC2 do not express precursor genes (line 127), they later state that precursor genes are downregulated (line 141). I understand what they are saying, but the text is a bit confusing.

d. The authors should confirm that CCR2 and CCR4 are expressed on the cell surface in their reporter mice.

e. The authors should be careful with statements about ILC2 trafficking (i.e. line 151) as they don't provide direct experimental evidence of this here.

f. Do homeostatic or IL-25 or IL-33 stimulated tissue express elevated levels of CCR2 or CCR4 ligands?

Figure 4: The authors show that administration of IL-33 induces the emergence of CCR2-RFP+ ILC2 in the small intestines at early time points, while there was a shift towards CCR4-nGr+ ILC2 over the following days. Adoptive transfer of lung ILC2 were able to populate the gut, but not the reverse; ILC2p also seeded the lungs before the gut, suggesting that passage through the lung environment precedes population of the gut.

a. Did the authors perfuse tissues or perform analysis of blood to exclude the contribution of circulating ILC2 in their tissue analysis?

b. The authors provide elegant lung labelling data, which further support the lung-gut trafficking hypothesis (Fig 4e); this method should allow for longer term analysis, and my question is if lung-derived ILC2 engraft and persist in the gut. Could the authors perform the same experiment and analyse the mice several weeks alter? Also, naïve

mice should be used as a control for background labelling (i.e. labelling at early time-points; moreover, if these controls confirm high selectivity for lung, then this system should be used to assess the role for lung-gut axis in homeostatic conditions.

c. *Ccr2* is also expressed by other immune cells, and the authors need to consider depletion of other immune cells. While ILC3 are not affected, the authors show high expression of *Ccr2* in ILC3 subsets and ILCreg. Could the authors comment?

Figure 5: The authors use *Ccr2*- and *Ccr4*-deficient mice to demonstrate that CCR2 is important for the generation of both lung and gut ILC2.

Figure 6: The authors focus on postnatal development of ILC2 in the lung and gut. The authors show that postnatal HDM treatment increased both lung and gut ILC2, which resulted in an amplified response to IL-33 challenge.

a. Figure 6c shows the emergence of *Ccr2*-RFP+*Ccr4*-mNG-/+ ILC2 in the gut on day 12. This does not prove that lung ILC2 translocate to the gut, as stated in line 239.

However, the authors' ID2-CreERT2 model may provide this evidence.

b. The transfer experiment (Fig 6d) is not clearly described.

c. The effect of HDM on increased gut inflammation is tenuous; the authors previously show that IL-33 causes the rapid emergence of *Ccr2*+ ILC2 in the gut (Figure 4), and it is not clear if the increased response in HDM-treated mice is due to increased lung ILC2 in these sensitized mice instead of dysfunction of resident gut ILC2.

Reviewer #2 ILC, scRNA, transcriptomics (Remarks to the Author):

In this manuscript Dr. Wang and colleagues postulated that group 2 innate lymphoid cells (ILC2s) matured along the lung-intestine axis i.e. lung ILC2s migrated to the gut to mature in this organ. The authors generated this hypothesis based on the pseudotime analysis of scRNA-seq data on lung and gut ILCs. scRNA-seq data indicated that majority of lung ILC2s are immature and express CCR2, the majority of gut ILC2s are mature and express CCR4. The results on enrichment of specific chemokine receptors on ILC2 subsets from the lung and gut are interesting and represent a strength of the manuscript. Having said that, It is important to emphasize that chemokine receptor expression is not fully specific to the organ; some lung ILC2s express CCR4 and some gut ILC2s express CCR2 (Fig 4a). Accordingly, some lung ILC2s are mature and some gut ILC2s are immature per pseudotime analysis (Fig 3b). These data (Fig 4a and Fig 3b) complicate interpretation of other results. In subsequent experiments, the authors showed that CCR2 expression was observed on ST2^{high}KLRG1^{low} ILC2s that responded to IL33, CCR4 expression was observed on ST2^{low}KLRG1^{high} that responded to IL25. The authors then perform a series of experiments to test their hypothesis on the maturation of ILC2s along the lung-gut axis and examine the roles of CCR2 and CCR4 in ILC2 migration and function.

Specific comments:

1. Insufficient evidence to support the central hypothesis on migration and maturation of ILC2 along the lung-gut axis. To test the hypothesis, the authors performed adoptive transfers of ILC2s isolated from the lung and gut (Fig 4c), utilized pulmonary delivery of 4-hydroxytamoxifen to activate expression of tdTomato in ILC2s (*Id2*-Cre/ERT2R *Rosa26*-STOP-tdTomato mice; Fig 4e) and depleted CCR2+ cells through DT injections into *Ccr2*-mNeonGreen-Cre *Rosa26*-STOP-DTR mice. None of these approaches is definitive. Adoptive transfers provided evidence that after injection into the bloodstream, lung ILC2 can home to the gut and gut ILC2s are unable to home to the lung. However, this experiment does not provide an answer to a critical question whether ILC2 can exit the lung. Furthermore, it is unclear whether transferred lung ILC2s mature after entering the gut. The investigators did not show any data on CCR2/CCR4 and markers of maturation. The experiment with pulmonary administration of 4-hydroxytamoxifen is hard to interpret. Expression of tdTomato in intestinal ILC2s may be due to systemic spread of 4-hydroxytamoxifen. The experiment using DT

treatment of *Ccr2-mNeonGreen-Cre Rosa26-STOP-DTR* mice is also hard to interpret. Absence of gut ILC2s may be due to deletion of local/gut-resident CCR2+ ILC2s (some gut ILC2s are CCR2+, see Fig 4a) that serve as precursors for gut CCR4+ ILC2s, or due to deletion of other (non-ILC2) CCR2+ cells that support gut ILC2s. Similar explanation (impairment/reduction of gut-resident CCR2+ ILC2s and/or CCR2+ non-ILCs) can be given to data on CCR2-/- mice (Fig 5b). Taken together, additional experiments are necessary to support the hypothesis.

2. In most of experiments, to stimulate ILC2s, the authors used cytokine injections. To bring disease relevance, at least some of the critical experiments need to be done using relevant environmental factors such as allergens or parasites.

3. In the current version of the manuscript, fluorescent protein reporter mice are the sole approaches to analyze expression of key molecules (CCR2, CCR4). To confirm these results, endogenous CCR2 and CCR4 proteins should be measured.

4. To study transcriptional consequences of CCR2 and CCR4 deletion, the authors performed RNAseq on ILC2s sorted from CCR2-/- and CCR4-/- mice. A better approach would be to use mice with ILC2-specific deletion of CCR2.

5. Minor: It is unclear how the CD127+ gate was set (no distinct populations are shown) in Supplementary Fig 1a. Please, show your control stains used to set the gate.

Reviewer #3 mucosal immunology, lung-gut axis (Remarks to the Author):

Maturation and specialization of group 2 innate lymphoid cells through the lung-gut axis.

Zhao et al.

The authors describe a global transcriptional characterization of ILC subsets at two distinct mucosal surfaces.

Using these findings, the authors identify that lung and gut ILC2s express distinct chemokine receptors. The authors then demonstrate the developmental relationship and responsiveness of these distinct subsets to external cytokine stimulation. Using adoptive transfer experiments the authors show that shortly after transfer, lung resident ILC2 give rise to gut ILC2s, but gut ILC2 fail to do the reverse. The authors track the development of ILC2 postnatally and conclude that the lung serves as maturation origin for gut ILC2s.

While the developed tools are very interesting, the concept is in contrast to other reports, their claims lack support and a needs a more thorough analysis.

1. The authors describe a list of gene that selectively mark ILCs within the lung or the small intestine. It would be interesting to see a few of these markers confirmed by flow cytometry. This would certainly strengthen the authors findings and their claim that these markers are useful to track ILCs based on their organ of origin. For example, a staining for CD36 in gut vs lung ILC3s would be interesting to see.

2. What is the status of CCR2 and CCR4 on ILC2s in other mucosal and non-mucosal tissues (spleen, liver, skin or adipose tissue)? Is it possible multiple other lung-tissue axes exist and the trajectory of development stated by the authors isn't as stringent as claimed?

3. Does treatment with IL-33 or IL-25 promote CCL2, CCL17 or CCL22 expression in gut or lung?

4. Does IL-33 treatment directly regulate CCR2 expression on ILC2s or does it mediate the accumulation of CCR2+ ILC2s? An in vitro stimulation/culture of CCR2- ILC2Ps or gut CCR2-ILC2s could help to answer this and would clarify the authors' statement on line 163 page 6 (IL-33 was able to induce RFP+163 ILC2s in the gut).

5. Regarding the adoptive transfer of ILC2s from IL-33 treated mice into

immunodeficient mice in Figure 4c. IL-33 treatment of lungs can lead to an increase in of ILC2s in the lung draining lymph nodes. What would be the status of these ILC2s, especially when these cells may also migrate to the lung? It is important to consider these and other sources of CCR2+ ILC2.

6. I disagree with the authors' claim that "lung ILC2s are less mature and undergo maturation through the lung-gut axis." This may be a sole bias introduced by the treatment with IL-33 and needs better clarification, especially when considering that IL-25 treatment appears to direct the opposite migratory pathway of ILC2s - from the gut to the lung (PMID: 29302015). The interpretations made by the authors need to be supported better especially when considering IL-25 treatment. Along these lines, immunodeficient mice injected with lung ILC2 or gut ILC2s should be analyze after more than 6 weeks. I would be surprised to see that gut ILC2s aren't able to generated lung ILCs. A better phenotypic characterization of lung and gut ILC2s should be provided for these transfers. For example, what is the status of KLRG1 or CCR2 vs CCR4 on these cells after transfer? Would this still happen if cells would have been isolated from naïve donor mice, without IL-33 or IL-25 injection?

7. It would be very supportive to see the experiment in Figure 5d performed with Ccr4-/- ILC2P.

8. Using their reporter lines, where would the distinct ILC2 subset be located within the lung or intestinal tract? Studies by Ari Molofsky indicated distinct niches for ILC2s. An analysis of the respective ILC2 population should be included, especially after IL-33 and IL-25 treatment.

9. In line with the authors' final figure, would Ccr2-/- or Ccr4-/- be less susceptible to HDM induced asthma? A histopathological assessment of these mice paired with clinical scoring could help to add disease associated implications to the authors' findings.

Minor:

1. It could be helpful to confirm absence of NK cells in extended figure 1b by showing Gzma/B and Eomes expression as an additional feature plot. Or showing Eomes and Gzma/B staining on ILCs post purification. Similarly, Eomes expression should be included as a violin plot in Figure 1d.

2. Could the authors provide a list of genes defining the tissue-specific pathways used for the Gene Ontology analysis in Figure 2 d-f? This would be helpful for the readers and should in parts reflect the data in Figure 2 a-c. This could be provided as an supplementary excel sheet.

3. The sentences on line 500ff should be reworked. There appears to be something wrong. "Two days after 4-OHT treatment, these mice were treated with i.p. injection of IL-33 and frequency of tdTomato+ ILC2s were calculated and show as means±SD. (f-g) Depletion of ILC2s in the lung and gut." etc. Some sentences appear out of context.

4. What is the IL-18R expression status of the distinct ILC2 populations?

Point-by-point response to the reviewers

Reviewer #1

This manuscript build on the emerging idea that ILC are strongly influence by their tissue environment. Extending on previously published ILC2 datasets, the authors now add both gut and lung ILC data (which nicely captures ILC1/2/3 and ILCreg). Remarkably, tissue-origin has a much stronger influence than ILC-lineage on the transcriptome. Bioinformatics analysis suggests that lung ILC2 serve as precursors for ILC2 in the gut, and that CCR2/CCR4 represent accurate markers for the lung and gut ILC2. Using adoptive transfer, detailed characterization, lineage trace and depletion experiments, the authors provide very compelling data that support a lung-to-gut developmental axis. One weakness is that the authors do not directly investigate the role of this axis in gut pathologies. The compelling, as other studies have suggested that ILC2 can locally arise from tissue-resident precursors (i.e. Zeis P 2020, Ghaedi M 2020). Also, given the long-term residence of ILC2 at baseline (i.e. Gasteiger G 2015), it remains uncertain how this mechanism contributes to gut ILC2 homeostasis over longer time periods.

Overall, this manuscript is thought provoking and of broad interest. The authors have generated or used elegant transgenic models, which can be used to address some of the outstanding questions (see below).

Answer: We appreciate the comments from this reviewer. We have tried our best to improve our manuscript according to the comments by reviewers.

Figure 1: The authors perform scRNAseq analysis on lung and gut enriched ILCs. They find the expected ILC subtypes.

a. Do the authors identify IL-18R+ ILC precursors in the lung, which are known to give rise to ILC2/1?

Answer: This is a very good point. We isolated the effector ILCs from the lung and the intestine. According to our data, the ILCs did not express Il18r1 (Attached Figure 1a), indicating that IL-18R+ ILC precursors were not involved in the ILC subsets in the scRNA data.

Figure 2: The authors perform more detailed analysis of the scRNAseq dataset, and mainly find that lung ILC express high levels of Ccr2 while gut ILC express more Ccr4.

Figure 3: Focussing on ILC2, the authors show by pseudotime analysis lung ILC2 may serve as a precursor for gut ILC2 subsets. The authors then used Ccr2 and Ccr4 reporter mice to validate their 10X dataset by flow cytometry analysis.

a. The gating strategy for ILC2 is not clear, it appears that the majority of lung ILC2 do not

express ST2 or KLRG1, which is i

Answer: We used the gate strategy as following: Lin⁻CD127⁺ST2⁺KLRG1⁺ for lung ILC2s at steady state, Lin⁻CD127⁺ST2⁻KLRG1⁺ for iILC2 upon treatment of IL-25, and Lin⁻CD127⁺KLRG1⁺ for gut ILC2s according to previous reports (PMID: 25531830). Majority of lung ILC2s were ST2⁺ at steady state or treated with IL-33. iILC2s induced by IL-25 were ST2⁻ as previously reported. We have described the gate strategy in the figure legends in our new version.

b. The authors propose that gut ILC2 comprise two rather distinct subgroups, which arise from a common precursor. What are the defining functional genes of the two gut subgroups (at their most mature end-states), and could the authors speculate about their function in homeostatic conditions?

Answer: According to our scRNA-seq data, ILC2-SI-B expressed higher levels of *Gata3*, *Klrg1*, *Il2ra*, *Cxcr6* (Attached Figure 1a), indicating that ILC2-SI-B is more mature than ILC2-SI-A subset. They might be affected by microbiota in the gut and regulate the intestine homeostasis.

c. While the authors state that lung ILC2 do not express precursor genes (line 127), they later state that precursor genes are downregulated (line 141). I understand what they are saying, but the text is a bit confusing.

Answer: We appreciate the comment from this reviewer. We modified our wording in the new version.

d. The authors should confirm that CCR2 and CCR4 are expressed on the cell surface in their reporter mice.

Answer: This is a very good point. We confirmed the expression of CCR2 and CCR4 in the reporter mice (Supplementary Figure 3a).

e. The authors should be careful with statements about ILC2 trafficking (i.e. line 151) as they don't provide direct experimental evidence of this here.

Answer: We appreciate the comment from this reviewer. We modified our wording in the new version.

f. Do homeostatic or IL-25 or IL-33 stimulated tissue express elevated levels of CCR2 or CCR4 ligands?

Answer: This is a very good point. We analyzed the expression levels of CCR2 and CCR4 ligand (CCL2, CCL17/CCL22 respectively) in the lung and gut. Their ligands were increased upon stimulation (Supplementary Figure 4a). Notably, CCL2 is significantly increased in the lung epithelial cells upon IL-33 treatment.

Figure 4: The authors show that administration of IL-33 induces the emergence of CCR2-RFP+ ILC2 in the small intestines at early time points, while there was a shift towards CCR4-nGr+ ILC2 over the following days. Adoptive transfer of lung ILC2 were able to populate the gut, but not the reverse; ILC2p also seeded the lungs before the gut, suggesting that passage through the lung environment precedes population of the gut.

a. Did the authors perfuse tissues or perform analysis of blood to exclude the contribution of circulating ILC2 in their tissue analysis?

Answer: This is a very good point. We perfused the lung and gut followed by flow cytometry analysis of ILC2s, and obtained the similar result. We also analyzed the blood ILC2s from and we did not observe CCR2⁺CCR4⁺ILC2s in the blood, indicating that CCR2⁺CCR4⁺ILC2s were not from the blood (Attached Figure 1b).

b. The authors provide elegant lung labelling data, which further support the lung-gut trafficking hypothesis (Fig 4e); this method should allow for longer term analysis, and my question is if lung-derived ILC2 engraft and persist in the gut. Could the authors perform the same experiment and analyse the mice several weeks alter? Also, naïve mice should be used as a control for background labelling (i.e. labelling at early time-points; moreover, if these controls confirm high selectivity for lung, then this system should be used to assess the role for lung-gut axis in homeostatic conditions.

Answer: This is a very good point. We labeled the Id2⁺ cells by using this model and analyzed the ILC2s three weeks after labelling. We only detected TdTomato⁺ILC2s in the gut upon IL-33 treatment, indicating the translocation of ILC2s after IL-33 stimulation (Attached Figure 2a).

c. Ccr2 is also expressed by other immune cells, and the authors need to consider depletion of other immune cells. While ILC3 are not affected, the authors show high expression of Ccr2 in ILC3 subsets and ILCreg. Could the authors comment?

Answer: This is a very good suggestion. Since monocytes express CCR2, we used *Rosa26-STOP-DTR;Lyz2-Cre* mouse model to deplete monocyte to see whether depletion of monocyte affect the ILC2 population . With the administration of DT, monocytes were successfully depleted (Supplementary Figure 3g), but the cell number of ILC2s were not affected (Supplementary Figure 3h), suggesting that depletion of CCR2⁺ monocytes did not affect ILC2 population. Although ILC3 and ILCreg express CCR2, we did not observe their cell number change in CCR2 deficient mice at steady state. CCR2 might play a regulatory role in them under inflammatory stimulation, which is worthy to be further analyzed.

Figure 5: The authors use Ccr2- and Ccr4-deficient mice to demonstrate that CCR2 is

important for the generation of both lung and gut ILC2.

Figure 6: The authors focus on postnatal development of ILC2 in the lung and gut. The authors show that postnatal HDM treatment increased both lung and gut ILC2, which resulted in an amplified response to IL-33 challenge.

a. Figure 6c shows the emergence of Ccr2-RFP+Ccr4-mNG-/+ ILC2 in the gut on day 12. This does not prove that lung ILC2 translocate to the gut, as stated in line 239. However, the authors' ID2-CreERT2 model may provide this evidence.s

Answer: We appreciate the comment from this reviewer. We intranasally administrated TMX carefully at postnatal day 12. The lung ILC2s were labelled, and then TdTomato⁺ILC2s appeared in the gut, indicating the lung-gut translocation of ILC2 at postnatal stage (Attached Figure 3f).

b. The transfer experiment (Fig 6d) is not clearly described.

Answer: We modified the description of the transfer experiment in our new version of manuscript.

c. The effect of HDM on increased gut inflammation is tenuous; the authors previously show that IL-33 causes the rapid emergence of Ccr2⁺ ILC2 in the gut (Figure 4), and it is not clear if the increased response in HDM-treated mice is due to increased lung ILC2 in these sensitized mice instead of dysfunction of resident gut ILC2.

Answer: This is a very good point. At postnatal day 12 of mice, few gut ILC2s were observed. But we could see accumulation of ILC2s in the lung, which is consistent with the results from Lambrecht's group (PMID: 27939673). Thus, we proposed that the effect of HDM was majorly on lung ILC2s upon intranasally treatment at that time point.

Reviewer #2 ILC, scRNA, transcriptomics (Remarks to the Author):

In this manuscript Dr. Wang and colleagues postulated that group 2 innate lymphoid cells (ILC2s) matured along the lung-intestine axis i.e. lung ILC2s migrated to the gut to mature in this organ. The authors generated this hypothesis based on the pseudotime analysis of scRNA-seq data on lung and gut ILCs. scRNA-seq data indicated that majority of lung ILC2s are immature and express CCR2, the majority of gut ILC2s are mature and express CCR4. The results on enrichment of specific chemokine receptors on ILC2 subsets from the lung and gut are interesting and represent a strength of the manuscript. Having said that, It is important to emphasize that chemokine receptor expression is not fully specific to the organ; some lung ILC2s express CCR4 and some gut ILC2s express CCR2 (Fig 4a). Accordingly, some lung ILC2s are mature and some gut ILC2s are immature per pseudotime analysis (Fig 3b). These data (Fig 4a and Fig 3b) complicate interpretation of

other results. In subsequent experiments, the authors showed that CCR2 expression was observed on ST2^{high}KLRG1^{low} ILC2s that responded to IL33, CCR4 expression was observed on ST2^{low}KLRG1^{high} that responded to IL25. The authors then perform a series of experiments to test their hypothesis on the maturation of ILC2s along the lung-gut axis and examine the roles of CCR2 and CCR4 in ILC2 migration and function.

Specific comments:

1. Insufficient evidence to support the central hypothesis on migration and maturation of ILC2 along the lung-gut axis. To test the hypothesis, the authors performed adoptive transfers of ILC2s isolated from the lung and gut (Fig 4c), utilized pulmonary delivery of 4-hydroxytamoxifen to activate expression of tdTomato in ILC2s (*Id2-Cre/ERT2R Rosa26-STOP-tdTomato* mice; Fig 4e) and depleted CCR2⁺ cells through DT injections into *Ccr2-mNeonGreen-Cre Rosa26-STOP-DTR* mice. None of these approaches is definitive. Adoptive transfers provided evidence that after injection into the bloodstream, lung ILC2 can home to the gut and gut ILC2s are unable to home to the lung. However, this experiment does not provide an answer to a critical question whether ILC2 can exit the lung. Furthermore, it is unclear whether transferred lung ILC2s mature after entering the gut. The investigators did not show any data on CCR2/CCR4 and markers of maturation. The experiment with pulmonary administration of 4-hydroxytamoxifen is hard to interpret. Expression of tdTomato in intestinal ILC2s may be due to systemic spread of 4-hydroxytamoxifen. The experiment using DT treatment of *Ccr2-mNeonGreen-Cre Rosa26-STOP-DTR* mice is also hard to interpret. Absence of gut ILC2s may be due to deletion of local/gut-resident CCR2⁺ ILC2s (some gut ILC2s are CCR2⁺, see Fig 4a) that serve as precursors for gut CCR4⁺ ILC2s, or due to deletion of other (non-ILC2) CCR2⁺ cells that support gut ILC2s. Similar explanation (impairment/reduction of gut-resident CCR2⁺ ILC2s and/or CCR2⁺ non-ILCs) can be given to data on CCR2^{-/-} mice (Fig 5b). Taken together, additional experiments are necessary to support the hypothesis.

Answer: We appreciate the comments from this reviewer. We have tried our best to improve our manuscript according to the reviewer's comments. We would like to explain and improve our findings from three major parts. **Firstly**, in order to label the ILCs in the lung, we used *Id2-ERT2-Cre;Rosa26-STOP-TdTomato* mouse model and 4-OHT (2µg/g mouse) were atomized and delivered into mouse lung by Liquid Aerosol Devices. According to previous study, the half-life of 4-OHT is only 0.5–2 h in vivo and suitable for transient labelling of cells in vivo (PMID: 12060754, PMID: 25996136). We have tested different concentration of 4-OHT and chosen the suitable dose that only labels the lung ILCs. We next analyzed the ILC2s in the lung and the gut at steady state several days after 4-OHT administration (Attached Figure 2b). Only lung ILC2s were labelled and express TdTomato, but gut ILC2s did not express TdTomato. Therefore, we supposed that this is a suitable model for the labelling of lung ILCs. When treated with IL-33, we can see the

TdTomato⁺ILC2s in the gut, indicating they might come from the lung.

Secondly, DT treatment of *Ccr2-mNeonGreen-Cre;Rosa26-STOP-DTR* mice abrogated ILC2 population in the lung and the gut. We used this model because using a lineage tracing system, we noticed that CCR2 was expressed in lung ILC2s and gut ILC2s used to be CCR2⁺ (Attached Figure 2c). Both lung and gut ILC2s were DTR positive (Attached Figure 2d). Therefore, the using of DT could deplete ILC2s in the lung and the gut (Figure 4f). In addition, according to the scRNA-sequencing data and qPCR data (Figure 3a and Supplementary Figure 3c), gut ILC2s did not express *Ccr2* mRNA. CCR2 was majorly expressed on lung ILC2s (Attached Figure 3e), and IL-33 treatment was not able to induce *Ccr2* expression in gut ILC2s in vitro and in vivo (Supplementary Figure 3d, Figure 4b). Thus, the CCR2⁺ ILC2s (actually RFP positive) in the gut upon IL-33 treatment might come from the lung. Next, we found that IL-33 induces the emergence of CCR2-RFP⁺ ILC2 in the small intestines at early time points, while there was a shift towards CCR4-mNG⁺ ILC2 over the following days, indicating the differentiation of CCR2-RFP⁺ ILC2 to CCR4-mNG⁺ ILC2 (Figure 4b). Moreover, by in vivo labelling experiment and adoptive transfer experiment, we found that adoptive transfer of lung ILC2 were able to populate the gut, but not the reverse. Thus, we hypothesize that ILC2s might undergo translocation from lung to the gut.

Thirdly, to exclude the effect of depletion of CCR2⁺ monocytes, we used *Rosa26-STOP-DTR;Lyz2-Cre* mouse model to deplete monocyte to see whether depletion of monocyte affect the ILC2 population. With the administration of DT, monocytes were successfully depleted (Supplementary Figure 3g), but the cell number of ILC2s were not affected, suggesting that depletion of CCR2⁺ monocytes did not affect ILC2 population (Supplementary Figure 3h).

2. In most of experiments, to stimulate ILC2s, the authors used cytokine injections. To bring disease relevance, at least some of the critical experiments need to be done using relevant environmental factors such as allergens or parasites.

Answer: This is a very good point. We used allergen house dust mite (HDM) to analyze the translocation of ILC2s from the lung to the gut. Similar results were obtained compared with cytokine injection (Attached Figure 2e, 2f).

3. In the current version of the manuscript, fluorescent protein reporter mice are the sole approaches to analyze expression of key molecules (CCR2, CCR4). To confirm these results, endogenous CCR2 and CCR4 proteins should be measured.

Answer: This is a very good point. We confirmed the expression of CCR2 and CCR4 in the reporter mice (Supplementary Figure 3a).

4. To study transcriptional consequences of CCR2 and CCR4 deletion, the authors performed RNAseq on ILC2s sorted from CCR2^{-/-} and CCR4^{-/-} mice. A better approach would be to use mice with ILC2-specific deletion of CCR2.

Answer: This is a very good point. However, all the known ILC2 related Cre mouse, for example *Id2-Cre*, *PLZF-Cre*, *Rora-Cre* are not ILC2 specific Cre. To analyze the intrinsic role of CCR2 and CCR4 on ILC2s, we transferred CCR2 or CCR4 deficient ILC2 precursors into NDG mice. ILC2s can be reconstituted successfully in these mice and we obtained the similar results of ILC2s in *Ccr2^{-/-}* and *Ccr4^{-/-}* mice upon cytokine stimulation (Attached Figure 3a, 3b). After treatment with IL-33, CCR2 deficient ILC2s were decrease. Moreover, CCR4 deficient ILC2s were accumulated in the lung upon treatment with IL-25. Therefore, we conclude that the effect of CCR2 and CCR4 is intrinsic on ILC2s.

5. Minor: It is unclear how the CD127⁺ gate was set (no distinct populations are shown) in Supplementary Fig 1a. Please, show your control stains used to set the gate.

Answer: We appreciate the comments from this reviewer. We added the control staining for this data (Attached Figure 3c).

Reviewer #3 mucosal immunology, lung-gut axis (Remarks to the Author):

Maturation and specialization of group 2 innate lymphoid cells through the lung-gut axis.

Zhao et al.

The authors describe a global transcriptional characterization of ILC subsets at two distinct mucosal surfaces.

Using these findings, the authors identify that lung and gut ILC2s express distinct chemokine receptors. The authors then demonstrate the developmental relationship and responsiveness of these distinct subsets to external cytokine stimulation. Using adoptive transfer experiments the authors show that shortly after transfer, lung resident ILC2 give rise to gut ILC2s, but gut ILC2 fail to do the reverse. The authors track the development of ILC2 postnatally and conclude that the lung serves as maturation origin for gut ILC2s.

While the developed tools are very interesting, the concept is in contrast to other reports, their claims lack support and a needs a more thorough analysis.

Answer: We appreciate the comments from this reviewer. We have tried our best to improve our manuscript according to the comments by reviewers.

1. The authors describe a list of gene that selectively mark ILCs within the lung or the small intestine. It would be interesting to see a few of these markers confirmed by flow cytometry. This would certainly strengthen the authors findings and their claim that these markers are

useful to track ILCs based on their organ of origin. For example, a staining for CD36 in gut vs lung ILC3s would be interesting to see.

Answer: This is a very good point. We stained CD36 and Ffar2 in ILC3s. We found that CD36 was specifically expressed on lung ILC3s and Ffar2 was mainly expressed on gut ILC3s (Attached Figure 3d)

2. What is the status of CCR2 and CCR4 on ILC2s in other mucosal and non-mucosal tissues (spleen, liver, skin or adipose tissue)? Is it possible multiple other lung-tissue axes exist and the trajectory of development stated by the authors isn't as stringent as claimed?

Answer: We appreciate the comments from this reviewer. We analyzed the expression of CCR2 and CCR4 on ILC2s in other tissues. At steady state, CCR2 was majorly expressed on lung ILC2s and CCR4 are more specifically expressed on gut ILC2s (Attached Figure 3e). It is possible that lung ILC2s may migrate to other tissues. According to the trajectory analysis of ILC2s in various tissues, lung ILC2s located at the initial position of the development trajectory and might give rise to more mature ILC2s in other tissues (Figure S2d). We found lung ILC2s were able to populate the gut, but not the reverse by using adoptive transfer assay (Figure 4c), lineage tracing model (Figure 4e), and reporter mice (Figure 4a, 4b). Whether other lung-tissue axes exist is worthy to be further investigated.

3. Does treatment with IL-33 or IL-25 promote CCL2, CCL17 or CCL22 expression in gut or lung?

Answer: This is a very good point. We analyzed the expression levels of CCR2 and CCR4 ligand (CCL2, CCL17/CCL22 respectively). CCL2 was significantly increased after IL-33 treatment in the lung, and CCL17 and CCL22 were increased by IL-25 treatment (Supplementary Figure 4a).

4. Does IL-33 treatment directly regulate CCR2 expression on ILC2s or does it mediate the accumulation of CCR2⁺ ILC2s? An in vitro stimulation/culture of CCR2⁻ ILC2Ps or gut CCR2-ILC2s could help to answer this and would clarify the authors' statement on line 163 page 6 (IL-33 was able to induce RFP⁺ ILC2s in the gut).

Answer: We appreciate the suggestion from this reviewer. We analyzed the mRNA level of CCR2 and found that IL-33 treatment did not affect the mRNA levels of ILC2s in the lung and gut (Supplementary Figure 4b). Moreover, we treated CCR2⁻ILC2Ps and gut ILC2s in vitro, we did not observe the induced expression of CCR2 (Supplementary Figure 3d). We observed the RFP⁺ILC2s in the gut upon IL-33 treatment (Figure 4c), but not the induced mRNA level of *Ccr2* gene (Supplementary Figure 3c), indicating that RFP⁺ILC2s in the gut just maintained the RFP protein from the lung.

5. Regarding the adoptive transfer of ILC2s from IL-33 treated mice into immunodeficient mice in Figure 4c. IL-33 treatment of lungs can lead to an increase in of ILC2s in the lung

draining lymph nodes. What would be the status of these ILC2s, especially when these cells may also migrate to the lung? It is important to consider these and other sources of CCR2⁺ ILC2.

Answer: This is a very good point. We analyzed the expression of CCR2 on ILC2s in other tissues. At steady state, CCR2 was majorly expressed on lung ILC2s (Attached Figure 3e), indicating that CCR2⁺ILC2s were mainly located in the lung. Upon stimulation, CCR2-RFP⁺ILC2s emerged in the gut, and there was a shift of towards CCR4-mNG⁺ ILC2s over the following days, indicating that ILC2s might come from the lung and differentiation into CCR4⁺ILC2s in the gut.

6. I disagree with the authors' claim that "lung ILC2s are less mature and undergo maturation through the lung-gut axis." This may be a sole bias introduced by the treatment with IL-33 and needs better clarification, especially when considering that IL-25 treatment appears to direct the opposite migratory pathway of ILC2s - from the gut to the lung (PMID: 29302015). The interpretations made by the authors need to be supported better especially when considering IL-25 treatment. Along these lines, immunodeficient mice injected with lung ILC2 or gut ILC2s should be analyze after more than 6 weeks. I would be surprised to see that gut ILC2s aren't able to generated lung ILCs. A better phenotypic characterization of lung and gut ILC2s should be provided for these transfers. For example, what is the status of KLRG1 or CCR2 vs CCR4 on these cells after transfer? Would this still happen if cells would have been isolated from naïve donor mice, without IL-33 or IL-25 injection?

Answer: We appreciate the suggestion from this reviewer. According to our scRNA-seq data, gut ILC2s expressed higher levels of *Gata3*, *Klrg1*, *Ii2ra*, which indicating the maturation of ILC2s (PMID: 23063333). Thus, we propose that gut ILC2s are more mature. Treatment with IL-25 or IL-33 indeed show different regulation mechanism. IL-25 treatment induces the migration of ILC2 from gut to the lung (PMID: 29302015). However, several reports show the IL-33 induced-migration of ILC2s. Locksley's group reported that during infection, second wave of ILC2s induced by IL-33 were from the lung and was abrogated in IL-33R-deficient mice (PMID: 32031571), indicating that IL-33 might induce the ILC2 migration from the lung. Moreover, IL-33 promotes the egress of ILC2s from BM (PMID: 32031571), and critically enhances ILC2 hematogenous migration and tissue repopulation in a lung-tropic manner, indicating that IL-33 might induce different regulation pathway from IL-25. IL-33 is closely related with ILC2 maturation and activation (PMID: 27939673). Our study mainly focusses on the regulation of IL-33 on ILC2s and showed the lung-gut axes regulated by IL-33. We also modified some of our wording in the new version.

According to the suggestion from this reviewer, we transferred NDG mice with lung or gut ILC2s. After 6 weeks, we analyzed the expression levels of *Klrg1*, *Ccr2* and *Ccr4* of ILC2s in lung-ILC2-transferred mice, the expression levels of *Klrg1* of gut ILC2s is much

higher than that in lung ILC2s (Attached Figure 4a). We also observed gut ILC2s from the lung-ILC2-transferred mice six weeks after transfer (Attached Figure 4b). When treated with IL-25, we observed iILC2 in the lung. It is possible that IL-25 induced the gut-lung transfer of ILC2s, and IL-33 induce lung-gut transfer of ILC2s.

7. It would be very supportive to see the experiment in Figure 5d performed with *Ccr4*^{-/-} ILC2P

Answer: To analyze the intrinsic role of CCR4 in ILC2s, we transferred CCR4 deficient ILC2 precursors into NDG mice. ILC2s can be reconstituted in these mice and we obtained the similar results of ILC2s in *Ccr4*^{-/-} mice (Attached Figure 3b). CCR4 deficient ILC2s were accumulated in the lung upon treatment with IL-25, similar with the results from CCR4 deficient mice. The development of ILC2s in BM is not affected in *Ccr4*^{-/-} mice.

8. Using their reporter lines, where would the distinct ILC2 subset be located within the lung or intestinal tract? Studies by Ari Molofsky indicated distinct niches for ILC2s. An analysis of the respective ILC2 population should be included, especially after IL-33 and IL-25 treatment.

Answer: We analyzed the distribution of ILC2s in the lung and the intestine (Attached Figure 4c) of CCR2-RFP;CCR4-mNG reporter mice. We observed the RFP⁺ILC2s in the lung and mNG⁺ILC2s in the gut. We also observed the RFP⁺mNG⁺ILC2s in the gut, which is consistent with the FACS data (Figure 4b).

9. In line with the authors' final figure, would *Ccr2*^{-/-} or *Ccr4*^{-/-} be less susceptible to HDM induced asthma? A histopathological assessment of these mice paired with clinical scoring could help to add disease associated implications to the authors' findings.

Answer: We analyzed the histopathological change of *Ccr2*^{-/-} or *Ccr4*^{-/-} mouse upon HDM treatment (Attached Figure 4d). We evaluated the peribronchial and perivascular lung inflammation score according to previously reported scoring system (PMID: 10606934), and found that *Ccr2*^{-/-} mice were less susceptible to HDM induced asthma. However, *Ccr4*^{-/-} mice showed similar pathological change with WT mice, indicating that CCR2 are more important for the function of ILC2s.

Minor:

1. It could be helpful to confirm absence of NK cells in extended figure 1b by showing GzmA/B and Eomes expression as an additional feature plot. Or showing Eomes and GzmA/B staining on ILCs post purification. Similarly, Eomes expression should be included as a violin plot in Figure 1d.

Answer: We analyzed the NK cell markers of the ILC subsets. *Eomes* are not expressed

by all the ILC subsets. *Gzma/b* were expressed by ILC1s, which is consistent with previous studies (PMID: 25621825, PMID: 34462601) (new Figure 1d and Supplementary Figure 1b).

2. Could the authors provide a list of genes defining the tissue-specific pathways used for the Gene Ontology analysis in Figure 2 d-f? This would be helpful for the readers and should in parts reflect the data in Figure 2 a-c. This could be provided as a supplementary excel sheet.

Answer: We provide the gene list of GO analysis in our new version (Supplementary Table 1).

3. The sentences on line 500ff should be reworked. There appears to be something wrong. "Two days after 4-OHT treatment, these mice were treated with i.p. injection of IL-33 and frequency of tdTomato+ ILC2s were calculated and show as means±SD. (f-g) Depletion of ILC2s in the lung and gut." etc. Some sentences appear out of context.

Answer: We appreciate the comment from this reviewer. We modified our wording in the new version.

4. What is the IL-18R expression status of the distinct ILC2 populations?

Answer: This is a very good point. We isolated the effector ILCs from the lung and the intestine. According to our data, the ILCs did not express *Il18r1* (Attached Figure 1a), indicating that IL-18R⁺ ILC precursors were not involved in the ILC subsets in the scRNA data.

Attached Figure 1-4

Attached Figure 1. (a) Expression levels of ILC signature genes were analyzed and shown in violin plot. (b) Analysis of ILC2s in blood and gut tissue of CCR2-RFP;CCR4-mNG reporter mice. CCR2-RFP;CCR4-mNG reporter mice were i.p. injected with 400 ng/mouse/day IL-33 for three constitutive days. The mice were sacrificed and perfused with 0.2% EDTA/PBS. The gut and blood ILC2s from the IL-33-treated mice were analyzed by flow cytometry. (ILC2=Lin⁻CD127⁺ KLRG1⁺, Lin=CD3e, CD19,CD11b,CD11c,Gr1,F4/80, NK1.1).PBS treatment served as a control (Ctrl).

Attached Figure 2. (a) Tracing of lung ILC2s. Solubilizing 4-OHT (2 μ g/g mouse) were atomized and delivered into lung of *Id2-Cre/ERT2;Rosa26-STOP-TdTomato* mice by Liquid Aerosol Devices as described in Materials and Methods. Three weeks after 4-OHT treatment, the mice were treated with or without IL-33 for two days, and gut ILC2s were analyzed by flow cytometry. The frequency of TdTomato⁺ ILC2s were calculated and show as means \pm SD. (b) Solubilizing 4-OHT (2 μ g/g mouse) were atomized and delivered into lung of *Id2-Cre/ERT2;Rosa26-STOP-TdTomato* mice by Liquid Aerosol Devices. After indicated days of 4-OHT treatment, lung and gut ILC2s were analyzed by flow cytometry and the frequency of TdTomato⁺ ILC2s were calculated and show as means \pm SD. (c) Lineage tracing of CCR2⁺ILC2s. Lung or gut ILC2s from *CCR2-mNeonGreen-Cre;Rosa26-STOP-TdTomato* mice were analyzed by flow cytometry. (d) Expression of diphtheria toxin receptor (HBEGF) on ILC2s. The lung and gut ILC2s from *CCR2-mNeonGreen-Cre;Rosa26-STOP-DTR* mice were stained with antibodies against ILC2 markers and DTR (HBEGF) and analyzed by flow cytometry. (e) Analysis of mNeonGreen and RFP expression of gut ILC2s from *CCR2-RFP;CCR4-mNeonGreen* mice after intranasal administration of 10 μ g HDM/mouse for the indicated days. The cell frequency of indicated subgroups of gut ILC2s was calculated and shown as means \pm SD. (n=3 for each group). (f) Solubilizing 4-OHT (2 μ g/g mouse) were atomized and delivered into lung of *Id2-Cre/ERT2;Rosa26-STOP-TdTomato* mice by Liquid Aerosol Devices. Two days after 4-OHT treatment, these mice were intranasal administration of 10 μ g HDM/mouse for the indicated days and frequency of TdTomato⁺ ILC2s were calculated and show as means \pm SD.

Attached Figure 3. (a) Adoptive transfer of WT and *Ccr2*^{-/-} ILC2Ps. ILC2Ps (Lin⁻CD127⁺Sca1⁺ST2⁺KLRG1⁻) were isolated from BM of wild type (WT) and *Ccr2*^{-/-} mice. WT and *Ccr2*^{-/-} ILC2Ps were 1:1 mixed (5x10⁴ for each) and i.v. injected into B-NDG mice. One week after transfer, the recipient mice were i.p. injected with 400 ng/mouse/day IL-33 for three days. The cell frequency of ILC2s in the lung and gut were calculated and shown as means±SD. (n=3 for each group). (b) Adoptive transfer of WT and *Ccr4*^{-/-} ILC2Ps. ILC2Ps (Lin⁻CD127⁺Sca1⁺ST2⁺KLRG1⁻) were isolated from BM of WT and *Ccr4*^{-/-} mice. WT and *Ccr4*^{-/-} ILC2Ps were 1:1 mixed (5x10⁴ for each) and i.v. injected into B-NDG mice. One week after transfer, the recipient mice were i.p. injected with 400 ng/mouse/day IL-33 or IL-25 for three days. The cell frequency of ILC2s in the lung and gut were calculated and shown as means±SD. (n=3 for each group). (c) Analysis of CD127 expression on ILCs. ILCs were isolated from the lung of WT mice and analyzed by flow cytometry. Fluorescence Minus One (FMO) showed as gate staining control of CD127. (d) Expression of CD36 and Ffar2 on the lung and gut IILC3s were analyzed by flow cytometry. Isotype IgG served as negative controls. (e) Expression of CCR2 and CCR4 on ILC2s. Expression of CCR2 and CCR4 on ILC2s from indicated tissues of CCR2-mNeonGreen reporter mice (upper panel) and CCR4-mNeonGreen reporter mice (lower panel) were analyzed by flow cytometry. WT mice served as a negative control. (f) Tracing of postnatal lung ILC2s. Solubilizing 4-OHT (1 µg/g mouse) were intranasally delivered into lung of mice at postnatal day 12. Three

days after administration, the lung and gut ILC2s were analyzed by flow cytometry. The frequency of TdTomato⁺ ILC2s were calculated and show as means±SD.

Attached Figure 4. (a-b) Adoptive transfer of lung or gut ILC2s (5×10^4) from WT CD45.1 mice to B-NDG mice. Six weeks after of transfer, the ILC2s from lung-ILC2 transferred mice were harvested for qPCR assay of indicated genes (a). Alternatively, the recipient mice were treated with or without 200ng/mouse IL-25 i.p. for one day. The lung and gut ILC2s from the recipient mice were analyzed by flow cytometry. The cell frequency of ILC2s were calculated and shown as means±SD. (b). The expression levels of *Klrg1* and *Ccr2* were normalized with that of lung ILC2s. The expression level of *Ccr4* were normalized with that of gut ILC2s. ND, not detectable. (n=3 for each group). (c) CCR2-RFP;CCR4-mNeonGreen mice were i.p. injected with IL-25 or IL-33. After 72 hours, the lung or gut tissues of mice were collected and fixed. The sections of lung and gut tissues from the treated mice were subjected to the immunostaining of indicated antibodies. Red arrowhead, RFP⁺ILC2; Blue arrowhead, mNG⁺ILC2; White arrowhead, RFP⁺mNG⁺ILC2. Scale bar, 100 μm. (d) CCR2 or CCR4 deficient mice were intranasally treated with 10 μg HDM/mouse for 5 consecutive days and the inflammation score were analyzed and shown as means±SD. (n=3 for each group).

Reviewer #1 (Remarks to the Author):

Dear authors,

thank you for the additional data/analysis, my concerns have been addressed adequately. My only comment is that the authors should try to integrate the requested data into their manuscript.

Reviewer #2 (Remarks to the Author):

The authors addressed my comments.

I have one additional suggestion to improve the manuscript. To address some of my previous comments, the authors presented additional figures in the rebuttal document ("Attached" Figures 2b, 2d, 2e, 2f and 3c). These figures should be included in the manuscript or its supplement.

Reviewer #3 (Remarks to the Author):

Reviewer #3 mucosal immunology, lung-gut axis (Remarks to the Author):

Maturation and specialization of group 2 innate lymphoid cells through the lung-gut axis.

Zhao et al.

The authors describe a global transcriptional characterization of ILC subsets at two distinct mucosal surfaces.

Using these findings, the authors identify that lung and gut ILC2s express distinct chemokine receptors. The authors then demonstrate the developmental relationship and responsiveness of these distinct subsets to external cytokine stimulation. Using adoptive transfer experiments the authors show that shortly after transfer, lung resident ILC2 give rise to gut ILC2s, but gut ILC2 fail to do the reverse. The authors track the development of ILC2 postnatally and conclude that the lung serves as maturation origin for gut ILC2s.

While the developed tools are very interesting, the concept is in contrast to other reports, their claims lack support and a needs a more thorough analysis.

Answer : We appreciate the comments from this reviewer. We have tried our best to improve our manuscript according to the comments by reviewers.

1. The authors describe a list of gene that selectively mark ILCs within the lung or the small intestine. It would be interesting to see a few of these markers confirmed by flow cytometry. This would certainly strengthen the authors findings and their claim that these markers are useful to track ILCs based on their organ of origin. For example, a staining for CD36 in gut vs lung ILC3s would be interesting to see.

Answer: This is a very good point. We stained CD36 and Ffar2 in ILC3s. We found that CD36 was specifically express on lung ILC3s and Ffar2 was mainly expressed on gut ILC3s (Attached Figure 3d)

R3: Thank you. This data needs to be included as supplemental figure in the manuscript.

2. What is the status of CCR2 and CCR4 on ILC2s in other mucosal and non-mucosal tissues (spleen, liver, skin or adipose tissue)? Is it possible multiple other lung-tissue axes exist and the trajectory of development stated by the authors isn't as stringent as claimed?

Answer: We appreciate the comments from this reviewer. We analyzed the expression of CCR2 and CCR4 on ILC2s in other tissues. At steady state, CCR2 was majorly expressed on lung ILC2s and CCR4 are more specifically expressed on gut ILC2s (Attached Figure 3e). It is possible that lung ILC2s may migrate to other tissues. According to the trajectory analysis of ILC2s in various tissues, lung ILC2s located at the initial position of the development trajectory and might give rise to more mature ILC2s in other tissues (Figure S2d). We found lung ILC2s were able to populate the gut, but not the reverse by using adoptive transfer assay (Figure 4c), lineage tracing model (Figure 4e), and reporter mice (Figure 4a, 4b). Whether other lung-tissue axes exist is worthy to be further investigated.

R3: Thank you. The data in attached figure 3e needs to be included as supplemental figure in the manuscript. A better quantification of the percentages of the respective ILC2 subsets within each tissue is needed. It appears that not only the gut, but the other tissue-resident ILC2 populations express CCR2. This challenges the authors' model of a lung-gut axis. Does the reconstitution pattern described by the authors lung first, then the gut also apply to the liver, skin and spleen? The authors new state the status of CCR2 or CCR4 on the injected ILC2Ps. This is another important consideration to make. Which route of i.v. injection did the authors choose for their adoptive transfers? Why did the authors inject ILC2Ps i.p. and not i.v.?

3. Does treatment with IL-33 or IL-25 promote CCL2, CCL17 or CCL22 expression in gut or lung?
Answer: This is a very good point. We analyzed the expression levels of CCR2 and CCR4 ligand (CCL2, CCL17/CCL22 respectively). CCL2 was significantly increased after IL-33 treatment in the lung, and CCL17 and CCL22 were increased by IL-25 treatment (Supplementary Figure 4a).

R3: Thank you. This data needs to be included as supplemental figure in the manuscript.

4. Does IL-33 treatment directly regulate CCR2 expression on ILC2s or does it mediate the accumulation of CCR2+ ILC2s? An in vitro stimulation/culture of CCR2- ILC2Ps or gut CCR2-ILC2s could help to answer this and would clarify the authors' statement on line 163 page 6 (IL-33 was able to induce RFP+ ILC2s in the gut).

Answer: We appreciate the suggestion from this reviewer. We analyzed the mRNA level of CCR2 and found that IL-33 treatment did not affect the mRNA levels of ILC2s in the lung and gut (Supplementary Figure 4b). Moreover, we treated CCR2-ILC2Ps and gut ILC2s in vitro, we did not observe the induced expression of CCR2 (Supplementary Figure 3d). We observed the RFP+ILC2s in the gut upon IL-33 treatment (Figure 4c), but not the induced mRNA level of Ccr2 gene (Supplementary Figure 3c), indicating that RFP+ILC2s in the gut just maintained the RFP protein from the lung.

R3: Thank you. This data needs to be included as supplemental figure in the manuscript.

5. Regarding the adoptive transfer of ILC2s from IL-33 treated mice into immunodeficient mice in Figure 4c. IL-33 treatment of lungs can lead to an increase in of ILC2s in the lung draining lymph nodes. What would be the status of these ILC2s, especially when these cells may also migrate to the lung? It is important to consider these and other sources of CCR2+ ILC2.

Answer: This is a very good point. We analyzed the expression of CCR2 on ILC2s in other tissues. At steady state, CCR2 was majorly expressed on lung ILC2s (Attached Figure 3e), indicating that CCR2+ILC2s were mainly located in the lung. Upon stimulation, CCR2-RFP+ILC2s emerged in the gut, and there was a shift of towards CCR4-mNG+ ILC2s over the following days, indicating that ILC2s might come from the lung and differentiation into CCR4+ILC2s in the gut.

6. I disagree with the authors' claim that "lung ILC2s are less mature and undergo maturation through the lung-gut axis." This may be a sole bias introduced by the treatment with IL-33 and needs better clarification, especially when considering that IL-25 treatment appears to direct the opposite migratory pathway of ILC2s - from the gut to the lung (PMID: 29302015). The interpretations made by the authors need to be supported better especially when considering IL-25 treatment. Along these lines, immunodeficient mice injected with lung ILC2 or gut ILC2s should be analyze after more than 6 weeks. I would be surprised to see that gut ILC2s aren't able to generated lung ILCs. A better phenotypic characterization of lung and gut ILC2s should be provided for these transfers. For example, what is the status of KLRG1 or CCR2 vs CCR4 on these cells after transfer? Would this still happen if cells would have been isolated from naïve donor mice, without IL-33 or IL-25 injection?

Answer: We appreciate the suggestion from this reviewer. According to our scRNA-seq data, gut ILC2s expressed higher levels of Gata3, Klrg1, Il2ra, which indicating the maturation of ILC2s (PMID: 23063333). Thus, we propose that gut ILC2s are more mature. Treatment with IL-25 or IL-33 indeed show different regulation mechanism. IL-25 treatment induces the migration of ILC2 from gut to the lung (PMID: 29302015). However, several reports show the IL-33 induced-migration of ILC2s. Locksley's group reported that during infection, second wave of ILC2s induced by IL-33 were from the lung and was abrogated in IL-33R-deficient mice (PMID: 32031571),

indicating that IL-33 might induce the ILC2 migration from the lung. Moreover, IL-33 promotes the egress of ILC2s from BM (PMID: 32031571), and critically enhances ILC2 hematogenous migration and tissue repopulation in a lung-tropic manner, indicating that IL-33 might induce different regulation pathway from IL-25. IL-33 is closely related with ILC2 maturation and activation (PMID: 27939673). Our study mainly focusses on the regulation of IL-33 on ILC2s and showed the lung-gut axes regulated by IL-33. We also modified some of our wording in the new version. According to the suggestion from this reviewer, we transferred NDG mice with lung or gut ILC2s. After 6 weeks, we analyzed the expression levels of Klrg1, Ccr2 and Ccr4 of ILC2s in lung-ILC2-transferred mice, the expression levels of Klrg1 of gut ILC2s is much higher than that in lung ILC2s (Attached Figure 4a). We also observed gut ILC2s from the lung-ILC2-transferred mice six weeks after transfer (Attached Figure 4b). When treated with IL-25, we observed iILC2 in the lung. It is possible that IL-25 induced the gut-lung transfer of ILC2s, and IL-33 induce lung-gut transfer of ILC2s.

R3: Thank you. This data needs to be included as supplemental figure in the manuscript. In the figure legend for attached figure 4. What day are the authors referring to in this sentence "Alternatively, the recipient mice were treated with or without 200ng/mouse IL-25 i.p. for one day." ? In attached Figure 4b. What is the statistical significance when comparing Lung ILC2 counts in untreated mice originating from the different organs. Would authors be able to perform a selective labelling of gut-resident ILC2s? The data as presented doesn't fully convince. The lung environment in the hosting B-NDG mice may not allow the engraftment of gut-derived ILC2s. I appreciate the efforts by the authors, but I don't think that their conclusion are fully supported by their results.

7. It would be very supportive to see the experiment in Figure 5d performed with Ccr4^{-/-} ILC2P
Answer: To analyze the intrinsic role of CCR4 in ILC2s, we transferred CCR4 deficient ILC2 precursors into NDG mice. ILC2s can be reconstituted in these mice and we obtained the similar results of ILC2s in Ccr4^{-/-} mice (Attached Figure 3b). CCR4 deficient ILC2s were accumulated in the lung upon treatment with IL-25, similar with the results from CCR4 deficient mice. The development of ILC2s in BM is not affected in Ccr4^{-/-} mice.

R3: Thank you. This data needs to be included as supplemental figure in the manuscript.

8. Using their reporter lines, where would the distinct ILC2 subset be located within the lung or intestinal tract? Studies by Ari Molofsky indicated distinct niches for ILC2s. An analysis of the respective ILC2 population should be included, especially after IL-33 and IL-25 treatment.
Answer: We analyzed the distribution of ILC2s in the lung and the intestine (Attached Figure 4c) of CCR2-RFP;CCR4-mNG reporter mice. We observed the RFP+ILC2s in the lung and mNG+ILC2s in the gut. We also observed the RFP+mNG+ILC2s in the gut, which is consistent with the FACS data (Figure 4b).

R3: Thank you. This data needs to be included as supplemental figure in the manuscript.

9. In line with the authors' final figure, would Ccr2^{-/-} or Ccr4^{-/-} be less susceptible to HDM induced asthma? A histopathological assessment of these mice paired with clinical scoring could help to add disease associated implications to the authors' findings.

Answer: We analyzed the histopathological change of Ccr2^{-/-} or Ccr4^{-/-} mouse upon HDM treatment (Attached Figure 4d). We evaluated the peribronchial and perivascular lung inflammation score according to previously reported scoring system (PMID: 10606934), and found that Ccr2^{-/-} mice were less susceptible to HDM induced asthma. However, Ccr4^{-/-} mice showed similar pathological change with WT mice, indicating that CCR2 are more important for the function of ILC2s.

R3: Thank you. This data needs to be included as supplemental figure in the manuscript and representative images of the scored tissues need to be shown.

Minor:

1. It could be helpful to confirm absence of NK cells in extended figure 1b by showing GzmA/B and Eomes expression as an additional feature plot. Or showing Eomes and GzmA/B staining on ILCs post purification. Similarly, Eomes expression should be included as a violin plot in Figure 1d.
Answer: We analyzed the NK cell markers of the ILC subsets. Eomes are not expressed by all the ILC subsets. Gzma/b were expressed by ILC1s, which is consistent with previous studies (PMID: 25621825, PMID: 34462601) (new Figure 1d and Supplementary Figure 1b).

R3: Thank you. Very good!

2. Could the authors provide a list of genes defining the tissue-specific pathways used for the Gene Ontology analysis in Figure 2 d-f? This would be helpful for the readers and should in parts reflect the data in Figure 2 a-c. This could be provided as a supplementary excel sheet.

Answer: We provide the gene list of GO analysis in our new version (Supplementary Table 1).

R3: Thank you. This data needs to be included as supplemental material in the manuscript.

3. The sentences on line 500ff should be reworked. There appears to be something wrong. "Two days after 4-OHT treatment, these mice were treated with i.p. injection of IL-33 and frequency of tdTomato+ ILC2s were calculated and show as means±SD. (f-g) Depletion of ILC2s in the lung and gut." etc. Some sentences appear out of context.

Answer: We appreciate the comment from this reviewer. We modified our wording in the new version.

R3: Thank you.

4. What is the IL-18R expression status of the distinct ILC2 populations?

Answer: This is a very good point. We isolated the effector ILCs from the lung and the intestine. According to our data, the ILCs did not express Il18r1 (Attached Figure 1a), indicating that IL-18R+ ILC precursors were not involved in the ILC subsets in the scRNA data.

R3: Thank you. This data needs to be included as supplemental figure in the manuscript. A detailed discussion of these findings need to be added to the manuscript to explain these contrasting findings in light of: PMID: 31816636 and PMID: 33002412

Point-by-point response to the reviewers

Reviewer #1

Dear authors,

thank you for the additional data/analysis, my concerns have been addressed adequately. My only comment is that the authors should try to integrate the requested data into their manuscript.

Answer: We appreciate the comments from this reviewer. We have integrated these data into the revised manuscript.

Reviewer #2

The authors addressed my comments.

I have one additional suggestion to improve the manuscript. To address some of my previous comments, the authors presented additional figures in the rebuttal document ("Attached" Figures 2b, 2d, 2e, 2f and 3c). These figures should be included in the manuscript or its supplement.

Answer: We appreciate the comments from this reviewer. We have integrated these data into the revised manuscript.

Reviewer #3

The authors describe a global transcriptional characterization of ILC subsets at two distinct mucosal surfaces.

Using these findings, the authors identify that lung and gut ILC2s express distinct chemokine receptors. The authors then demonstrate the developmental relationship and responsiveness of these distinct subsets to external cytokine stimulation. Using adoptive transfer experiments the authors show that shortly after transfer, lung resident ILC2 give rise to gut ILC2s, but gut ILC2 fail to do the reverse. The authors track the development of ILC2 postnatally and conclude that the lung serves as maturation origin for gut ILC2s.

While the developed tools are very interesting, the concept is in contrast to other reports, their claims lack support and a needs a more thorough analysis.

Answer: We appreciate the comments from this reviewer. We have tried our best to improve our manuscript according to the comments by reviewers.

1. The authors describe a list of gene that selectively mark ILCs within the lung or the small intestine. It would be interesting to see a few of these markers confirmed by flow cytometry. This would certainly strengthen the authors findings and their claim that these markers are useful to track ILCs based on their organ of origin. For example, a staining for CD36 in gut vs lung ILC3s would be interesting to see.

Answer: This is a very good point. We stained CD36 and Ffar2 in ILC3s. We found that CD36 was specifically express on lung ILC3s and Ffar2 was mainly expressed on gut ILC3s (Attached Figure 3d)

R3: Thank you. This data needs to be included as supplemental figure in the manuscript.

Answer: We appreciate the comments from this reviewer. We have integrated these data into the revised manuscript.

2. What is the status of CCR2 and CCR4 on ILC2s in other mucosal and non-mucosal

tissues (spleen, liver, skin or adipose tissue)? Is it possible multiple other lung-tissue axes exist and the trajectory of development stated by the authors isn't as stringent as claimed?
Answer: We appreciate the comments from this reviewer. We analyzed the expression of CCR2 and CCR4 on ILC2s in other tissues. At steady state, CCR2 was majorly expressed on lung ILC2s and CCR4 are more specifically expressed on gut ILC2s (Attached Figure 3e). It is possible that lung ILC2s may migrate to other tissues. According to the trajectory analysis of ILC2s in various tissues, lung ILC2s located at the initial position of the development trajectory and might give rise to more mature ILC2s in other tissues (Figure S2d). We found lung ILC2s were able to populate the gut, but not the reverse by using adoptive transfer assay (Figure 4c), lineage tracing model (Figure 4e), and reporter mice (Figure 4a, 4b). Whether other lung-tissue axes exist is worthy to be further investigated.

R3: Thank you. The data in attached figure 3e needs to be included as supplemental figure in the manuscript. A better quantification of the percentages of the respective ILC2 subsets within each tissue is needed. It appears that not only the gut, but the other tissue-resident ILC2 populations express CCR2. This challenges the authors' model of a lung-gut axis. Does the reconstitution pattern described by the authors lung first, then the gut also apply to the liver, skin and spleen? The authors new state the status of CCR2 or CCR4 on the injected ILC2Ps. This is another important consideration to make. Which route of i.v. injection did the authors choose for their adoptive transfers? Why did the authors inject ILC2Ps i.p. and not i.v.?

Answer: We have integrated these data into the revised manuscript. We also show the frequency of the indicated ILC2 subsets in the new version (Supplementary Figure 3a). Actually, gut ILC2s do not express CCR2, and the mRNA of *Ccr2* was not observed in the gut ILC2s according to our scRNA-seq data (Figure 3a) and qPCR analysis (Supplementary Figure 3f). CCR2-mNG⁺ILC2s were observed in the gut upon IL-33 treatment because of the translocation of these cells that maintained CCR2-mNG protein. It is possible that gut ILC2s translocate to other organs upon different stimulation, which is worthy to be further investigated. We observed that ILC2s underwent lung-gut translocation upon IL-33 treatment. We intravenously injected ILC2Ps into B-NDG mice through the tail vein of mice. Then we intraperitoneally injected the mice with IL-33. We described the details in the figure legends.

3. Does treatment with IL-33 or IL-25 promote CCL2, CCL17 or CCL22 expression in gut or lung?

Answer: This is a very good point. We analyzed the expression levels of CCR2 and CCR4 ligand (CCL2, CCL17/CCL22 respectively). CCL2 was significantly increased after IL-33 treatment in the lung, and CCL17 and CCL22 were increased by IL-25 treatment (Supplementary Figure 4a).

R3: Thank you. This data needs to be included as supplemental figure in the manuscript.

Answer: We have integrated these data into the revised manuscript.

4. Does IL-33 treatment directly regulate CCR2 expression on ILC2s or does it mediate the accumulation of CCR2+ ILC2s? An in vitro stimulation/culture of CCR2- ILC2Ps or gut CCR2-ILC2s could help to answer this and would clarify the authors' statement on line 163 page 6 (IL-33 was able to induce RFP+ ILC2s in the gut).

Answer: We appreciate the suggestion from this reviewer. We analyzed the mRNA level of CCR2 and found that IL-33 treatment did not affect the mRNA levels of ILC2s in the lung and gut (Supplementary Figure 4b). Moreover, we treated CCR2-ILC2Ps and gut ILC2s in vitro, we did not observe the induced expression of CCR2 (Supplementary Figure 3d). We observed the RFP+ILC2s in the gut upon IL-33 treatment (Figure 4c), but not the induced mRNA level of Ccr2 gene (Supplementary Figure 3c), indicating that RFP+ILC2s in the gut just maintained the RFP protein from the lung.

R3: Thank you. This data needs to be included as supplemental figure in the manuscript.

Answer: We have integrated these data into the revised manuscript.

5. Regarding the adoptive transfer of ILC2s from IL-33 treated mice into immunodeficient mice in Figure 4c. IL-33 treatment of lungs can lead to an increase in of ILC2s in the lung draining lymph nodes. What would be the status of these ILC2s, especially when these cells may also migrate to the lung? It is important to consider these and other sources of CCR2+ ILC2.

Answer: This is a very good point. We analyzed the expression of CCR2 on ILC2s in other tissues. At steady state, CCR2 was majorly expressed on lung ILC2s (Attached Figure 3e), indicating that CCR2+ILC2s were mainly located in the lung. Upon stimulation, CCR2-RFP+ILC2s emerged in the gut, and there was a shift of towards CCR4-mNG+ ILC2s over the following days, indicating that ILC2s might come from the lung and differentiation into CCR4+ILC2s in the gut.

6. I disagree with the authors' claim that "lung ILC2s are less mature and undergo maturation through the lung-gut axis." This may be a sole bias introduced by the treatment with IL-33 and needs better clarification, especially when considering that IL-25 treatment appears to direct the opposite migratory pathway of ILC2s - from the gut to the lung (PMID: 29302015). The interpretations made by the authors need to be supported better especially when considering IL-25 treatment. Along these lines, immunodeficient mice injected with lung ILC2 or gut ILC2s should be analyze after more than 6 weeks. I would be surprised to see that gut ILC2s aren't able to generated lung ILCs. A better phenotypic characterization of lung and gut ILC2s should be provided for these transfers. For example, what is the status of KLRG1 or CCR2 vs CCR4 on these cells after transfer? Would this still happen if cells would have been isolated from naïve donor mice, without IL-33 or IL-25 injection?

Answer: We appreciate the suggestion from this reviewer. According to our scRNA-seq data, gut ILC2s expressed higher levels of Gata3, Klrg1, Il2ra, which indicating the maturation of ILC2s (PMID: 23063333). Thus, we propose that gut ILC2s are more mature. Treatment with IL-25 or IL-33 indeed show different regulation mechanism. IL-25 treatment induces the migration of ILC2 from gut to the lung (PMID: 29302015). However, several

reports show the IL-33 induced-migration of ILC2s. Locksley's group reported that during infection, second wave of ILC2s induced by IL-33 were from the lung and was abrogated in IL-33R-deficient mice (PMID: 32031571), indicating that IL-33 might induce the ILC2 migration from the lung. Moreover, IL-33 promotes the egress of ILC2s from BM (PMID: 32031571), and critically enhances ILC2 hematogenous migration and tissue repopulation in a lung-tropic manner, indicating that IL-33 might induce different regulation pathway from IL-25. IL-33 is closely related with ILC2 maturation and activation (PMID: 27939673). Our study mainly focusses on the regulation of IL-33 on ILC2s and showed the lung-gut axes regulated by IL-33. We also modified some of our wording in the new version.

According to the suggestion from this reviewer, we transferred NDG mice with lung or gut ILC2s. After 6 weeks, we analyzed the expression levels of *Klrg1*, *Ccr2* and *Ccr4* of ILC2s in lung-ILC2-transferred mice, the expression levels of *Klrg1* of gut ILC2s is much higher than that in lung ILC2s (Attached Figure 4a). We also observed gut ILC2s from the lung-ILC2-transferred mice six weeks after transfer (Attached Figure 4b). When treated with IL-25, we observed iILC2 in the lung. It is possible that IL-25 induced the gut-lung transfer of ILC2s, and IL-33 induce lung-gut transfer of ILC2s.

R3: Thank you. This data needs to be included as supplemental figure in the manuscript. In the figure legend for attached figure 4. What day are the authors referring to in this sentence "Alternatively, the recipient mice were treated with or without 200ng/mouse IL-25 i.p. for one day." ? In attached Figure 4b. What is the statistical significance when comparing Lung ILC2 counts in untreated mice originating from the different organs. Would authors be able to perform a selective labelling of gut-resident ILC2s? The data as presented doesn't fully convince. The lung environment in the hosting B-NDG mice may not allow the engraftment of gut-derived ILC2s. I appreciate the efforts by the authors, but I don't think that their conclusion are fully supported by their results.

Answer: We have integrated these data into the revised manuscript. Six weeks after of transfer of lung or gut ILC2s (5×10^4) from WT CD45.1 mice to B-NDG mice, the recipient mice were treated with or without 200ng/mouse IL-25 i.p. for one day followed by further analysis. The transfer of lung or gut ILC2s was used to prove that lung ILC2s were able to populate the gut, but not the reverse (Figure 4c). We also used other assays to prove the exist of lung-gut axis of ILC2 translocation.

Firstly, in order to label the ILCs in the lung, we used *Id2-ERT2-Cre;Rosa26-STOP-TdTomato* mouse model and 4-OHT were atomized and delivered into mouse lung by Liquid Aerosol Devices. According to previous study, the half-life of 4-OHT is only 0.5–2 h in vivo and suitable for transient labelling of cells in vivo (PMID: 12060754, PMID: 25996136). Only lung ILC2s were labelled and express TdTomato, but gut ILC2s did not express TdTomato (Supplementary Figure 3j). Therefore, we supposed that this is a suitable model for the labelling of lung ILCs. When treated with IL-33, we can see the TdTomato⁺ILC2s in the gut, indicating they might come from the lung (Figure 4e).

Secondly, DT treatment of *Ccr2-mNeonGreen-Cre;Rosa26-STOP-DTR* mice abrogated ILC2 population in the lung and the gut. We used this model because using a lineage tracing system, we noticed that CCR2 was expressed in lung ILC2s and gut ILC2s

used to be CCR2⁺ (Supplementary Figure 4a). Both lung and gut ILC2s were DTR positive (Supplementary Figure 4b). Therefore, the using of DT could deplete ILC2s in the lung and the gut (Figure 4f). Next, we used IL-33 to promote the development and maturation of ILC2Ps from BM for the reconstruction of lung and gut ILC2s. Consistent with the adoptive transfer data, IL-33 administration induced the reconstruction of ILC2s first in the lung followed by the gut (Figure 4h).

Thirdly, according to the scRNA-seq data and qPCR data (Figure 3a and Supplementary Figure 3f), gut ILC2s did not express *Ccr2* mRNA. CCR2 was majorly expressed on lung ILC2s (Supplementary Figure 3a), and IL-33 treatment was not able to induce *Ccr2* expression in gut ILC2s in vitro and in vivo (Supplementary Figure 3g and Supplementary Figure 5b). Thus, the CCR2⁺ ILC2s (actually RFP positive) in the gut upon IL-33 treatment might come from the lung (Figure 4a). Next, we found that IL-33 induces the emergence of CCR2-RFP⁺ ILC2 in the small intestines at early time points, while there was a shift towards CCR4-mNG⁺ ILC2 over the following days, indicating the differentiation of CCR2-RFP⁺ ILC2 to CCR4-mNG⁺ ILC2 (Figure 4b).

Moreover, CCR4 was specifically expressed on gut ILC2s according to the scRNA-seq data and flow cytometry data (Figure 3a and Supplementary Figure 3a). We used CCR4-mNG reporter mice for the tracing of gut ILC2s. We did not observe CCR4-mNG⁺ILC2s in the gut at steady state or upon IL-33 treatment (Figure 3d and 3h), indicating the gut ILC2s did not translocated to the lung.

Collectively, apart from adoptive transfer assay, we used in vivo labelling and depletion experiments and cell tracing mouse models to analyze the translocation of ILC2s. We found that ILC2s might undergo translocation from lung to the gut upon IL-33 and HDM treatment.

7. It would be very supportive to see the experiment in Figure 5d performed with *Ccr4*^{-/-} ILC2P

Answer: To analyze the intrinsic role of CCR4 in ILC2s, we transferred CCR4 deficient ILC2 precursors into NDG mice. ILC2s can be reconstituted in these mice and we obtained the similar results of ILC2s in *Ccr4*^{-/-} mice (Attached Figure 3b). CCR4 deficient ILC2s were accumulated in the lung upon treatment with IL-25, similar with the results from CCR4 deficient mice. The development of ILC2s in BM is not affected in *Ccr4*^{-/-} mice.

R3: Thank you. This data needs to be included as supplemental figure in the manuscript.

Answer: We have integrated these data into the revised manuscript.

8. Using their reporter lines, where would the distinct ILC2 subset be located within the lung or intestinal tract? Studies by Ari Molofsky indicated distinct niches for ILC2s. An analysis of the respective ILC2 population should be included, especially after IL-33 and IL-25 treatment.

Answer: We analyzed the distribution of ILC2s in the lung and the intestine (Attached Figure 4c) of CCR2-RFP;CCR4-mNG reporter mice. We observed the RFP+ILC2s in the lung and mNG+ILC2s in the gut. We also observed the RFP+mNG+ILC2s in the gut, which is consistent with the FACS data (Figure 4b).

R3: Thank you. This data needs to be included as supplemental figure in the manuscript.

Answer: We have integrated these data into the revised manuscript.

9. In line with the authors' final figure, would Ccr2^{-/-} or Ccr4^{-/-} be less susceptible to HDM induced asthma? A histopathological assessment of these mice paired with clinical scoring could help to add disease associated implications to the authors' findings.

Answer: We analyzed the histopathological change of Ccr2^{-/-} or Ccr4^{-/-} mouse upon HDM treatment (Attached Figure 4d). We evaluated the peribronchial and perivascular lung inflammation score according to previously reported scoring system (PMID: 10606934), and found that Ccr2^{-/-} mice were less susceptible to HDM induced asthma. However, Ccr4^{-/-} mice showed similar pathological change with WT mice, indicating that CCR2 are more important for the function of ILC2s.

R3: Thank you. This data needs to be included as supplemental figure in the manuscript and representative images of the scored tissues need to be shown.

Answer: We have integrated these data into the revised manuscript. We showed the representative images of the scored tissues in our new version (Supplementary Figure 5e)

Minor:

1. It could be helpful to confirm absence of NK cells in extended figure 1b by showing GzmA/B and Eomes expression as an additional feature plot. Or showing Eomes and GzmA/B staining on ILCs post purification. Similarly, Eomes expression should be included as a violin plot in Figure 1d.

Answer: We analyzed the NK cell markers of the ILC subsets. Eomes are not expressed by all the ILC subsets. Gzma/b were expressed by ILC1s, which is consistent with previous studies (PMID: 25621825, PMID: 34462601) (new Figure 1d and Supplementary Figure 1b).

R3: Thank you. Very good!

2. Could the authors provide a list of genes defining the tissue-specific pathways used for the Gene Ontology analysis in Figure 2 d-f? This would be helpful for the readers and should in parts reflect the data in Figure 2 a-c. This could be provided as a supplementary excel sheet.

Answer: We provide the gene list of GO analysis in our new version (Supplementary Table 1).

R3: Thank you. This data needs to be included as supplemental material in the manuscript.

Answer: We have integrated these data into the revised manuscript.

3. The sentences on line 500ff should be reworked. There appears to be something wrong. "Two days after 4-OHT treatment, these mice were treated with i.p. injection of IL-33 and frequency of tdTomato+ ILC2s were calculated and show as means±SD. (f-g) Depletion of ILC2s in the lung and gut." etc. Some sentences appear out of context.

Answer: We appreciate the comment from this reviewer. We modified our wording in the new version.

R3: Thank you.

4. What is the IL-18R expression status of the distinct ILC2 populations?

Answer: This is a very good point. We isolated the effector ILCs from the lung and the intestine. According to our data, the ILCs did not express Il18r1 (Attached Figure 1a), indicating that IL-18R+ ILC precursors were not involved in the ILC subsets in the scRNA data.

R3: Thank you. This data needs to be included as supplemental figure in the manuscript. A detailed discussion of these findings need to be added to the manuscript to explain these contrasting findings in light of: PMID: 31816636 and PMID: 33002412

Answer: We have integrated these data into the revised manuscript. We have added the discussion of these findings in our new version

Reviewer #3 (Remarks to the Author):

6. I disagree with the authors' claim that "lung ILC2s are less mature and undergo maturation through the lung-gut axis." This may be a sole bias introduced by the treatment with IL-33 and needs better clarification, especially when considering that IL-25 treatment appears to direct the opposite migratory pathway of ILC2s - from the gut to the lung (PMID: 29302015). The interpretations made by the authors need to be supported better especially when considering IL-25 treatment. Along these lines, immunodeficient mice injected with lung ILC2 or gut ILC2s should be analyzed after more than 6 weeks. I would be surprised to see that gut ILC2s aren't able to generate lung ILCs. A better phenotypic characterization of lung and gut ILC2s should be provided for these transfers. For example, what is the status of KLRG1 or CCR2 vs CCR4 on these cells after transfer? Would this still happen if cells would have been isolated from naïve donor mice, without IL-33 or IL-25 injection?

Answer: We appreciate the suggestion from this reviewer. According to our scRNA-seq data, gut ILC2s expressed higher levels of *Gata3*, *Klrg1*, *Ii2ra*, which indicating the maturation of ILC2s (PMID: 23063333). Thus, we propose that gut ILC2s are more mature. Treatment with IL-25 or IL-33 indeed show different regulation mechanism. IL-25 treatment induces the migration of ILC2 from gut to the lung (PMID: 29302015). However, several reports show the IL-33 induced-migration of ILC2s. Locksley's group reported that during infection, second wave of ILC2s induced by IL-33 were from the lung and was abrogated in IL-33R-deficient mice (PMID: 32031571), indicating that IL-33 might induce the ILC2 migration from the lung. Moreover, IL-33 promotes the egress of ILC2s from BM (PMID: 32031571), and critically enhances ILC2 hematogenous migration and tissue repopulation in a lung-tropic manner, indicating that IL-33 might induce different regulation pathway from IL-25. IL-33 is closely related with ILC2 maturation and activation (PMID: 27939673). Our study mainly focusses on the regulation of IL-33 on ILC2s and showed the lung-gut axes regulated by IL-33. We also modified some of our wording in the new version.

According to the suggestion from this reviewer, we transferred NDG mice with lung or gut ILC2s. After 6 weeks, we analyzed the expression levels of *Klrg1*, *Ccr2* and *Ccr4* of ILC2s in lung-ILC2-transferred mice, the expression levels of *Klrg1* of gut ILC2s is much higher than that in lung ILC2s (Attached Figure 4a). We also observed gut ILC2s from the lung-ILC2-transferred mice six weeks after transfer (Attached Figure 4b). When treated with IL-25, we observed iILC2 in the lung. It is possible that IL-25 induced the gut-lung transfer of ILC2s, and IL-33 induce lung-gut transfer of ILC2s.

Thank you the reply. The possibility of each cytokines acting different on ILC2 should be thoroughly discussed and raised.

Point-by-point response to the reviewers

Reviewer #3

6. I disagree with the authors' claim that "lung ILC2s are less mature and undergo maturation through the lung-gut axis." This may be a sole bias introduced by the treatment with IL-33 and needs better clarification, especially when considering that IL-25 treatment appears to direct the opposite migratory pathway of ILC2s - from the gut to the lung (PMID: 29302015). The interpretations made by the authors need to be supported better especially when considering IL-25 treatment. Along these lines, immunodeficient mice injected with lung ILC2 or gut ILC2s should be analyzed after more than 6 weeks. I would be surprised to see that gut ILC2s aren't able to generate lung ILCs. A better phenotypic characterization of lung and gut ILC2s should be provided for these transfers. For example, what is the status of KLRG1 or CCR2 vs CCR4 on these cells after transfer? Would this still happen if cells would have been isolated from naïve donor mice, without IL-33 or IL-25 injection?

Answer: We appreciate the suggestion from this reviewer. According to our scRNA-seq data, gut ILC2s expressed higher levels of *Gata3*, *Klrg1*, *Il2ra*, which indicating the maturation of ILC2s (PMID: 23063333). Thus, we propose that gut ILC2s are more mature. Treatment with IL-25 or IL-33 indeed show different regulation mechanism. IL-25 treatment induces the migration of ILC2 from gut to the lung (PMID: 29302015). However, several reports show the IL-33 induced-migration of ILC2s. Locksley's group reported that during infection, second wave of ILC2s induced by IL-33 were from the lung and was abrogated in IL-33R-deficient mice (PMID: 32031571), indicating that IL-33 might induce the ILC2 migration from the lung. Moreover, IL-33 promotes the egress of ILC2s from BM (PMID: 32031571), and critically enhances ILC2 hematogenous migration and tissue repopulation in a lung-tropic manner, indicating that IL-33 might induce different regulation pathway from IL-25. IL-33 is closely related with ILC2 maturation and activation (PMID: 27939673). Our study mainly focusses on the regulation of IL-33 on ILC2s and showed the lung-gut axes regulated by IL-33. We also modified some of our wording in the new version.

According to the suggestion from this reviewer, we transferred NDG mice with lung or gut ILC2s. After 6 weeks, we analyzed the expression levels of *Klrg1*, *Ccr2* and *Ccr4* of ILC2s in lung-ILC2-transferred mice, the expression levels of *Klrg1* of gut ILC2s is much higher than that in lung ILC2s (Attached Figure 4a). We also observed gut ILC2s from the lung-ILC2-transferred mice six weeks after transfer (Attached Figure 4b). When treated with IL-25, we observed iILC2 in the lung. It is possible that IL-25 induced the gut-lung transfer of ILC2s, and IL-33 induce lung-gut transfer of ILC2s.

Thank you the reply. The possibility of each cytokines acting different on ILC2 should be thoroughly discussed and raised.

Answer: We have discussed and raised this in our revised version.